# OPTIMAL TRANSPORT FOR REDUCING BIAS IN CAUSAL INFERENCE WITHOUT DATA SPLITTING

## ABSTRACT

Causal inference seeks to estimate the causal effect given a treatment such as a kind of medicine or the dosage of a medication. To address the issue of confounding bias caused by the non-randomized treatment assignment on samples, most existing methods reduce the covariate shift between subpopulations receiving different values of treatment. However, these methods split training samples into smaller groups, which cuts down the number of samples in each group, while precise distribution estimation and alignment highly rely on a sufficient number of training data. In this paper, we propose a distribution alignment paradigm that involves all the training samples without data splitting, which can be naturally applied in the settings of binary and continuous treatments. To this end, we characterize the distribution shift by considering different probability measures of the same set including all the training samples, and reduce the shift between the marginal covariate distribution and the conditional covariate distribution given a treatment value. By doing this, data reduction caused by splitting is avoided, and the outcome prediction model trained on samples receiving one treatment value can be generalized to the entire population. In specific, we exploit the optimal transport theory built on probability measures to analyze the confounding bias and the outcome estimation error, which motivates us to propose a balanced representation learning method for causal inference of binary and continuous treatments. The experimental results on both binary and continuous treatment settings demonstrate the effectiveness of the proposed method.

## 1 INTRODUCTION

Causal inference aims to estimate the causal effects of treatments for supporting decision-making, where the treatments are usually binary (Shalit et al., 2017) or continuous (Schwab et al., 2020). The gold standard for estimating causal effects is to conduct randomized control trials (RCTs) (Fisher, 1936), in which the assignment of treatment for samples is completely random without relying on the covariates of samples. However, it is usually infeasible to conduct RCTs, and the effects are estimated from observational data involving confounding bias, which means that the data distribution of a subpopulation receiving one value of treatment differs from the distribution of the entire population (Hammerton & Munafò, 2021), *i.e.*, $p(x|t) \neq p(x)$, where $x$ is the covariates and $t$ is the treatment value.

To address the confounding bias, most existing methods adopt a data-splitting strategy to partition samples into smaller subpopulations according to the treatment values, and then reduce the distribution shift between different subpopulations. For binary treatments, one usually splits training samples to the treated group receiving treatment and the control group without receiving treatment, and then reduces the distribution shift between the two groups (Kuang et al., 2017; Shalit et al., 2017). For continuous treatments, the natural and widely used strategy is to split samples into multiple groups based on their received treatments. After that, the distribution shift reduction approach for binary treatments can be applied by considering the shift between each pair of groups (Wang et al., 2022). However, data splitting cuts down the number of samples in each subpopulation, and only a part of the samples are leveraged in distribution estimation and alignment. This decreases the performance of distribution estimation and confounding bias reduction, which highly relies on a sufficient number of training samples.

In this paper, we propose a distribution alignment paradigm involving all the training samples without data splitting, which can be naturally applied to effect estimation of binary and continuous treatments. Rather than reducing the distribution shift between subpopulations receiving different treatment values in existing methods, we characterize the distribution shift by different probability measures of the same set including all the samples. In other words, we model the conditional distribution $p(x|t)$ by all the samples, instead of only a subpopulation receiving $t$ which is widely used in existing works (Shalit et al., 2017; Wang et al., 2022). By doing this, data splitting is avoided and all the samples can be leveraged to improve the performance of distribution alignment.

In specific, we establish the connection between the treatment effect estimation and optimal transport built on probability measures involving all the samples (Villani, 2008; Peyré & Cuturi, 2017). We show that for the marginal covariate distribution and the conditional covariate distribution given a treatment value, both the bias of covariates and the bias of outcome estimation errors can be upper bounded by the Wasserstein distances between these two distributions. Motivated by our theoretical results, we propose a method named **O**ptimal transport for **R**educing b**I**as in **C**ausal inference (ORIC), which learns balanced representations to reduce the confounding bias and outcome estimation error jointly. As a result, the outcome prediction model trained on samples receiving one treatment value can be generalized to the entire population. Our theoretical results and algorithm can be naturally applied to both binary and continuous treatments. We conduct experiments on synthetic and semi-synthetic datasets under the binary and continuous treatment settings, and the results demonstrate the effectiveness of our proposed method compared with existing methods.

The principal contributions are summarized as follows:

- To address the confounding bias in causal inference, we propose to characterize the distribution shift by considering different probability measures of all the training samples without data splitting.
- We construct the theoretical connection between the estimation error of treatment outcomes and optimal transport, which measures the distribution shift between the marginal covariate distribution and the conditional covariate distribution given a treatment value.
- Motivated by our theoretical results, we propose a balanced representation learning algorithm to reduce confounding bias and outcome estimation error jointly, and conduct experiments under different settings to demonstrate the effectiveness of the method.

## 2 RELATED WORKS

### 2.1 CAUSAL EFFECT ESTIMATION

Causal inference has been widely used in real-world applications, such as economics (Davis & Heller, 2020; Kreif et al., 2021; Cockx et al., 2023), healthcare (Sanchez et al., 2022; Karboub & Tabaa, 2022; Van Goethem et al., 2021), and advertising (Chen et al., 2020; Liu et al., 2021; Wei et al., 2021). Due to the confounding bias, the data distribution of a subpopulation receiving one value of treatment differs from the distribution of the entire population (Hammerton & Munafò, 2021). For example, in the treatment of a disease, the group receiving surgery usually has more severe conditions compared with the group receiving medication, the patients receiving higher doses of drugs usually have more severe conditions compared with the patients receiving lower doses, resulting in a distribution discrepancy between a subpopulation and the entire population.

Most existing works consider the binary and continuous treatment settings. The binary setting only considers whether the treatment is conducted or not (Shalit et al., 2017; Shi et al., 2019; Zhang et al., 2020), and the continuous treatment setting considers the outcome of the dosage of the treatment to estimate the dose-response function (Schwab et al., 2020; Nie et al., 2021; Wang et al., 2022).

**Binary Treatment.** Causal effect estimation of binary treatments considers only two groups, *i.e.*, the one receiving the treatment the one not receiving the treatment (Chipman et al., 2010; Dismuke & Lindrooth, 2006; Yoon et al., 2018; Zhang et al., 2020). To address the confounding bias between the two groups, one class of methods is to create a pseudo-balanced group by learning weights for samples. Kuang et al. (2017) proposed to reweight samples by reducing the distribution discrepancy between the two groups, where the discrepancy is measured by the difference of the moments. The other class of methods is to learn balanced representations for the two groups

(Johansson et al., 2016). Shalit et al. (2017); Johansson et al. (2022) proposed to learn representations with the minimized distribution discrepancy between two groups, where the discrepancy is measured by the integral probability metric and a theoretical analysis regarding the effect estimation error is provided.

Our proposed learning model can be naturally applied in the binary treatment setting. Actually, distribution alignment between two groups split training samples into two subsets, also cutting down the number of samples in each group. By modeling a distribution as a probability measure of all the samples, we avoid data splitting and obtain more samples for learning.

**Continuous Treatment.** Causal effect estimation of continuous treatments considers that the treatment lies in an interval, *e.g.*, the dosage of a medication (Imbens, 2000). The natural strategy is to partition training samples into multiple groups, each of which receives a similar dose of the treatment. By doing this, the existing methods for binary treatments can be applied. Schwab et al. (2020) adopted a multi-head architecture to deal with multiple intervals of treatment separately. Wang et al. (2022) calculated the discrepancy between each pair of two groups and reduced the largest discrepancy to learn balanced representations. The strategy of data splitting cuts down the training samples in each group, highly affecting the performance of distribution estimation and alignment. Different from them, we characterize the distribution discrepancy by different probability measures of all the samples, avoiding data reduction in splitting.

There are also a few works of continuous treatments without data splitting. Nie et al. (2021) proposed a varying coefficient model to estimate the effects of continuous treatment and apply a targeted regularization paradigm for estimation. Different with it, we explicitly reduce the confounding bias and theoretically reveal the connection between the confounding bias and the generalization error of the outcome estimation, which are missing in (Nie et al., 2021). Kazemi & Ester (2024) measured the distribution discrepancy based on the Kullback-Leibler (KL) divergence and employed an adversarial learning paradigm to learn the representations. However, the KL divergence suffers from the issue of gradient vanish when the distribution discrepancy is too large (Arjovsky et al., 2017), and the adversarial architecture is usually difficult to train (Gulrajani et al., 2017). Different from it, we measure the discrepancy by the Wasserstein distance to avoid the issue of gradient vanish, which can be easily estimated by the Sinkhorn algorithm (Cuturi, 2013).

## 2.2 OPTIMAL TRANSPORT

Optimal transport studies how to move mass from one distribution to another with a minimal transport cost (Monge, 1781; Kantorovitch, 1958; Villani, 2008). Beneficial from the powerful ability to model probability distributions and exploit geometry, optimal transport has been widely applied in many applications (Peyré & Cuturi, 2017), such as computer vision (Rubner et al., 2000), domain adaptation (Courty et al., 2014; 2017), data generation (Arjovsky et al., 2017; Tolstikhin et al., 2018), graph data analysis (Peyré et al., 2016; Titouan et al., 2019), *etc*.

Optimal transport has also been introduced into causal effect estimation of binary treatments recently (Yan et al., 2024a;b; Wang et al., 2024). Li et al. (2021) proposed to transport the factual distribution to the counterfactual distribution for estimating counterfactual outcomes. Dunipace (2021) employed optimal transport to learn an intermediate distribution by reweighting samples. Different from the above studies that only consider binary treatments, we address the confounding bias in the setting of continuous treatments. Besides, in the above methods, a distribution usually considers only a subpopulation, while our model represents a distribution by involving all the training samples and a probability measure, improving the number of training samples for distribution estimation and alignment.

## 3 PROBLEM STATEMENT

We assume a dataset of the form $\{(x_i, t_i, y_i)\}_{i=1}^n$, where $(x, t, y)$ is a realization of random vector $(X, T, Y)$. Here $x_i \in \mathcal{X}$ denotes the covariates of the $i$-th sample, $t_i \in \mathcal{T}$ is the treatment value that the sample $i$ received which can be binary or continuous, and $y_i \in \mathcal{Y}$ denotes the outcome of interest for the sample $i$ after receiving treatment $t_i$. Under Neyman-Rubin potential outcome framework (Rubin, 1974; Rosenbaum & Rubin, 1983), the observed outcome $Y$ is the potential outcome $Y(t)$ corresponding to the actually received treatment $T = t$.

Given input covariates $X = x$ and the treatment $T = t$, our goal is to derive an estimator $h(x, t)$ for the ground-truth individual response function $\mu(x, t)$ as follow:

$$\mu(x, t) = \mathbb{E}[Y(t)|X = x]. \tag{1}$$

For simplicity, we will use the shorthand $\mu_t(x) = \mu(x, t)$ and $h_t(x) = h(x, t)$. The following assumptions have been made to ensure that $\mu_t(x)$ is identifiable from observational data.

**Assumption 1 (Stable Unit Treatment Value Assumption)** *The potential outcomes for any sample do not vary with the treatments assigned to other samples, and for each sample, there are no different forms or versions of each treatment value which leads to different potential outcomes.*

**Assumption 2 (Ignorability)** *Conditional on covariates, the treatment assignment is independent of potential outcomes: $T \perp\!\!\!\perp Y(t)|X$.*

**Assumption 3 (Positivity)** *Conditional on covariates, the treatment assignment is not deterministic: $0 < p(T = t|X = x) < 1$.*

With these assumptions, $\mu_t(x)$ can be rewritten as follows, and we can estimate it as :

$$\mu_t(x) = \mathbb{E}[Y(t)|X = x] = \mathbb{E}[Y|X = x, T = t]. \tag{2}$$

Without ambiguity, we omit the random variables to write $p(X = x)$ as $p(x)$ for simplicity.

## 4 METHODOLOGY

In this section, we first characterize the confounding bias by considering different probability measures of all the samples, in which data will not be split into subpopulations. After that, we provide theoretical results regarding the confounding bias and the generalization error of the outcome estimation from the perspective of optimal transport, which is built on probability measures of all the samples. Based on the theoretical analysis, we propose a balanced representation learning algorithm to reduce the confounding bias and outcome estimation error jointly.

### 4.1 CONFOUNDING BIAS IN CAUSAL EFFECT ESTIMATION

Given the set of Radon measures $\mathcal{M}(\mathcal{X})$, let the marginal covariate distribution be the probability measure $q \in \mathcal{M}(\mathcal{X})$, and the conditional covariate distribution given a treatment value $t \in \mathcal{T}$ be the probability measure $q_t \in \mathcal{M}(\mathcal{X})$. The corresponding probability density functions can be written as $q(x) = p(x)$, $q_t(x) = p(x|t)$. According to Assumption 3, for each sample $x$ and treatment value $t$, we have $q_t(x) = p(x|t) = p(x)p(t|x)/p(t) > 0$, which means all the samples could be drawn from the distribution $q_t$. Motivated by this, we model $q_t$ as a probability measure involving all the training samples, which is different from data splitting that only samples receiving $t$ are involved (Shalit et al., 2017; Wang et al., 2022).

In specific, for the treatment value $t$, based on the loss function $\ell : \mathcal{Y} \times \mathcal{Y} \to \mathbb{R}^+$, we aim to minimize the following estimation error on the marginal distribution $q(x)$

$$\varepsilon_q(h_t) = \varepsilon_q(h_t, \mu_t) = \mathbb{E}_{x \sim q}\ell(h_t(x), \mu_t(x)) = \int_{\mathcal{X}} \ell(h_t(x), \mu_t(x))q(x)dx, \tag{3}$$

and achieve a small average mean squared error (AMSE) considering all the possible values of treatment which is defined as

$$AMSE = \mathbb{E}_{t \sim p(t)}\varepsilon_q(h_t) = \int_{\mathcal{T}} \varepsilon_q(h_t)p(t)dt. \tag{4}$$

Nevertheless, given the observational data, we can only minimize the following factual error on the conditional distribution $q_t(x)$

$$\varepsilon_{q_t}(h_t) = \varepsilon_{q_t}(h_t, \mu_t) = \mathbb{E}_{x \sim q_t(x)}\ell(h_t(x), \mu_t(x)) = \int_{\mathcal{X}} \ell(h_t(x), \mu_t(x))q_t(x)dx. \tag{5}$$

The principal challenge in causal effect estimation comes from the confounding bias, *i.e.*, $q(x) \neq q_t(x), \forall t \in \mathcal{T}$. As a result, the model trained to minimize $\varepsilon_{q_t}$ cannot be well generalized to minimize $\varepsilon_q$. To measure the level of confounding bias between $q_t(x)$ and $q(x)$, given a function (*e.g.*, balancing score) $m(\cdot)$ and a norm $\| \cdot \|$, we define the balancing error between these two distributions as

$$\xi(m, t) = \|\mathbb{E}_{x \sim q_t(x)} m(x) - \mathbb{E}_{x \sim q(x)} m(x)\|$$
$$= \Big\| \int_{\mathcal{X}} q_t(x) m(x) dx - \int_{\mathcal{X}} q(x) m(x) dx \Big\|. \tag{6}$$

We consider all the possible treatment values $t \in \mathcal{T}$, and define the total balancing error as follows

$$\xi(m) = \int_{\mathcal{T}} \xi(m, t) p(t) dt$$
$$= \int_{\mathcal{T}} \Big\| \int_{\mathcal{X}} q_t(x) m(x) dx - \int_{\mathcal{X}} q(x) m(x) dx \Big\| p(t) dt. \tag{7}$$

We do not restrict the specific form of the function $m(\cdot)$ as long as it can capture information from samples, enabling the balancing error $\xi(\cdot)$ to characterize the degree of confounding bias.

In the following, we establish the connection between the treatment effect estimation and optimal transport, which motivates us to propose a balanced representation learning algorithm for reducing confounding bias and outcome estimation error.

### 4.2 THEORETICAL ANALYSIS

To analyze the confounding bias and outcome estimation error, we exploit the theory of optimal transport built on probability measures. Optimal transport aims to find the optimal plan to move mass from one distribution to another with a minimal transport cost (Villani, 2008; Peyré & Cuturi, 2017). Formally, for the samples from two spaces $\alpha \in \mathcal{A}$, $\beta \in \mathcal{B}$, let $\mathcal{M}(\mathcal{A})$ and $\mathcal{M}(\mathcal{B})$ be the sets of Radon measures. Consider two distributions $\alpha \in \mathcal{M}(\mathcal{A})$, $\beta \in \mathcal{M}(\mathcal{B})$, and a distance function $c : \mathcal{A} \times \mathcal{B} \to \mathbb{R}^+$ with the corresponding norm $\| \cdot \|$, the Wasserstein distance between two distributions $\mathcal{W}(c, \alpha, \beta)$ is defined by the following Kantorovich Problem

$$\mathcal{W}(c, \alpha, \beta) = KP(\alpha, \beta) = \inf_{\pi \in \Pi(\alpha, \beta)} \int_{\mathcal{A} \times \mathcal{B}} c(a, b) d\pi(a, b), \tag{8}$$

where $\pi$ is a transport plan, and $\Pi(\alpha, \beta)$ is the set of all joint probability couplings whose marginal distributions are $\alpha$ and $\beta$, respectively. $\pi(a, b)$ indicates how many masses are moved from $\alpha$ to $\beta$, and the transport cost between them is measured by the distance $c(a, b)$. The minimized transport cost calculated by the optimal plan is the Wasserstein distance to measure the discrepancy between two distributions.

Given the pair of continuous functions $(f, g)$ satisfying the constraint $f(a) + g(b) \leq c(a, b)$, the above Kantorovich problem admits the following Dual Problem (Villani, 2021)

$$DP(\alpha, \beta) = \sup_{f(a) + g(b) \leq c(a,b)} \int_{\mathcal{A}} f(a) d\alpha(a) + \int_{\mathcal{B}} g(b) d\beta(b). \tag{9}$$

The following theorem shows that the confounding bias can be upper bounded by the Wasserstein distances between the marginal covariate distribution and the conditional covariate distributions given a value of treatment.

**Theorem 1** *Let $q$ be the marginal covariate distribution, and $q_t$ be the conditional covariate distribution given the treatment value $t$, i.e., $q(x) = p(x)$ and $q_t(x) = p(x|t)$. Given a pair of the functions $(m, c)$ satisfying the condition $m(x_i) - m(x_j) \leq c(x_i, x_j)$. We have the following result*

$$\xi(m) \leq \int_{\mathcal{T}} \mathcal{W}(c, q_t, q) p(t) dt. \tag{10}$$

This theorem presents that the confounding bias characterized by the balancing error can be upper bounded by the Wasserstein distances based on an underlying cost function $c(\cdot, \cdot)$ and the probability

measures $q_t$ and $q$, where the cost function $c(\cdot, \cdot)$ can be implemented by a distance measured on a representation space.

However, only focusing on confounding bias reduction may lead to a trivial solution that loses outcome information, hampering the performance of outcome prediction. For the outcome estimation error, we can only train a prediction model $h_t$ on the training data to minimize $\varepsilon_{q_t}(h_t)$ in Eq. (5), while the objective is to minimize $\varepsilon_q(h_t)$ in Eq. (3). The bias of the outcome estimation errors $\varepsilon_{q_t}(h_t)$ and $\varepsilon_q(h_t)$ is characterized by the following theorem

**Theorem 2** *Assume that the cost function $c(x, x') = \|\phi(x) - \phi(x')\|_{\mathcal{H}}$, where $\mathcal{H}$ is a Reproducing Kernel Hilbert Space (RKHS) induced by $\phi : \mathcal{X} \to \mathcal{H}$. Assume further that $h_t, \mu_t \in \mathcal{F}$ where $\mathcal{F}$ is a unit ball in the RKHS $\mathcal{H}$, and the loss function $\ell(h_t(x), \mu_t(x))$ is convex, symmetric, bounded, obeys the triangular inequality and has the parametric form $|h_t(x) - \mu_t(x)|^\chi$ for some $\chi > 0$. Assume also that kernel $k$ in the RKHS $\mathcal{H}$ is square-root integrable with respect to $\mathcal{X}$ and $0 \le k(x, x') = \langle \phi(x), \phi(x') \rangle \le K$. Then the following holds.*

$$\int_{\mathcal{T}} \varepsilon_q(h_t)p(t)dt - \int_{\mathcal{T}} \varepsilon_{q_t}(h_t)p(t)dt \le \int_{\mathcal{T}} \mathcal{W}(c, q_t, q)p(t)dt. \tag{11}$$

This theorem shows that given an outcome prediction model $h_t$, the Wasserstein distances between the distributions $q$ and $q_t$ provide an upper bound for the bias between the outcome estimation errors of $h_t$ on $q$ and $q_t$. The theorem also indicates that it is not sufficient to reduce $\mathcal{W}(c, q_t, q)$ only, since a small $\mathcal{W}(c, q_t, q)$ cannot guarantee to obtain a model $h_t$ with good performance. Even a model $h_t$ with poor prediction performance can perform similarly on $q_t$ and $q$, which happens when the information about the outcome is missing during distribution alignment. Therefore, in order to minimize $AMSE$ that is the estimation error on $q$ defined in Eq. (4), we propose to minimize the estimation error on the conditional distributions $q_t$ and the Wasserstein distances between $q$ and $q_t$ simultaneously, as shown in the following

$$AMSE = \int_{\mathcal{T}} \varepsilon_q(h_t)p(t)dt \le \int_{\mathcal{T}} \varepsilon_{q_t}(h_t)p(t)dt + \int_{\mathcal{T}} \mathcal{W}(c, q_t, q)p(t)dt, \tag{12}$$

which can be obtained from Eq. (11) immediately.

For the probability measures $q_t$ and $q$, a convenient property of optimal transport is that either continuous or discrete measures can be handled within the same framework, and the probabilities $q_t(x)$ and $q(x)$ can be easily represented as the sample weights for empirical distributions (Peyré & Cuturi, 2017). In practice, given training samples $\{x_i\}_{i=1}^n$, let $\delta_{x_i}$ be the Dirac function at the location $x_i$, $\hat{q}_t(x_i)$ and $\hat{q}(x_i)$ are the probability masses of the sample $x_i$ in the distributions $q_t$ and $q$, respectively, which satisfy the simplex constraints

$$\sum_{i=1}^n \hat{q}_t(x_i) = 1, \qquad \sum_{i=1}^n \hat{q}(x_i) = 1. \tag{13}$$

The corresponding empirical distributions $\hat{q}_t$ and $\hat{q}$ can be represented as

$$\hat{q}_t = \sum_{i=1}^n \hat{q}_t(x_i)\delta_{x_i}, \qquad \hat{q} = \sum_{i=1}^n \hat{q}(x_i)\delta_{x_i}. \tag{14}$$

Here, all the training samples are involved in the empirical distributions, which avoids the issue of data splitting and enhances the performance of distribution estimation.

Based on this, the relation between the outcome estimation error and the Wasserstein distances measured on the empirical discrete distributions is provided in the following theorem.

**Theorem 3** *Let $n$ be the number of samples, $\hat{q}, \hat{q}_t$ be the empirical distributions of $q, q_t$, respectively. With the probability of at least $1 - \delta$, we have:*

$$AMSE \le \int_{\mathcal{T}} \varepsilon_{q_t}(h_t)p(t)dt + \int_{\mathcal{T}} \mathcal{W}(c, \hat{q}_t, \hat{q})p(t)dt + \mathcal{O}\left(1/\sqrt{\delta n}\right). \tag{15}$$

### 4.3 ALGORITHM

According to the above theoretical analysis, we propose to minimize the outcome prediction error on the observational distribution $q_t$ and the Wasserstein distances between the empirical distributions $\hat{q}_t$ and $\hat{q}$ with $t \in \mathcal{T}$. The first part of the right side of Inequality (15) is defined as

$$
\begin{aligned}
\mathcal{L} &= \int_{\mathcal{T}} \varepsilon_{q_t}(h_t) p(t) dt \\
&= \int_{\mathcal{X} \times \mathcal{T}} \ell(h_t(x), \mu_t(x)) p(t) p(x|t) dx dt \\
&= \int_{\mathcal{X} \times \mathcal{T}} \ell(h_t(x), \mu_t(x)) p(x, t) dx dt.
\end{aligned}
\tag{16}
$$

By implementing the hypothesis as $h_t(x) = \psi(\phi(x), t)$, where $\phi(\cdot)$ is a model for representation learning, and $\psi(\cdot)$ is for outcome prediction, the above loss can be written based on the empirical distribution of training samples by the following

$$
\widehat{\mathcal{L}} = \frac{1}{n} \sum_{i=1}^{n} \left( y_i - \psi(\phi(x_i), t_i) \right)^2.
\tag{17}
$$

The second part of the right side of Inequality (15) is to minimize the Wasserstein distances on the empirical distributions $\mathcal{W}(c, \hat{q}_t, \hat{q})$, where the cost function is measured in the embedding space, *i.e.*, $c(x_i, x_j) = c_\phi(x_i, x_j) = \|\phi(x_i) - \phi(x_j)\|$, and the Wasserstein distance is estimated by the following

$$
\mathcal{W}(c_\phi, \hat{q}_t, \hat{q}) = \sum_{i=1}^{n} \sum_{j=1}^{n} c_\phi(x_i, x_j) \tilde{\pi}_{ij}^t,
\tag{18}
$$

where $\tilde{\pi}^t$ is the solution of the following optimization problem

$$
\tilde{\pi}^t = \arg \min_{\pi^t \in \Pi^t} \; \sum_{i=1}^{n} \sum_{j=1}^{n} c_\phi(x_i, x_j) \pi_{ij}^t + \gamma \Omega(\pi^t)
$$

$$
\text{s.t. } \Pi^t = \{ \pi^t \in \mathbb{R}_+^{n \times n} \mid \sum_{j=1}^{n} \pi_{ij}^t = \hat{q}_t(x_i) \; \forall \, i, \sum_{i=1}^{n} \pi_{ij}^t = \hat{q}(x_j) \; \forall \, j \},
\tag{19}
$$

where the entropic regularization $\Omega(\pi^t) = \sum_{i=1}^{n} \sum_{j=1}^{n} \pi_{ij}^t \log \pi_{ij}^t$ is the negative entropy, $\gamma$ is the trade-off hyper-parameter, and the Sinkhorn algorithm can be applied to solve the problem efficiently (Cuturi, 2013).

The probability mass $\hat{q}(x_i)$ is approximated as $\frac{1}{n}$. For the probability mass $\hat{q}_t(x_i)$, since $q_t(x_i) = p(x_i|t) = \frac{p(x_i)}{p(t)} p(t|x_i) \propto p(t|x_i)$, we approximate $p(t|x_i)$ by $\hat{p}(t|x_i) = \theta(\phi(x_i))$, which is estimated by the generalized propensity score (Imbens, 2000) based on the model $\theta(\cdot)$. As a result, $\hat{q}_t(x_i)$ is approximated by the normalized value $\hat{q}_t(x_i) = \frac{1}{Z} \theta(\phi(x_i))$, where $Z = \sum_{i=1}^{n} \theta(\phi(x_i))$ is the normalized factor, so that the simplex constraint in Eq. (13) is satisfied.

In practice, similar to $\hat{q}_t$ and $\hat{q}$ that only consider the empirical discrete samples, we consider a set $\widehat{\mathcal{T}}$ including discrete values of the treatment. For binary treatments, we have $\widehat{\mathcal{T}} = \{0, 1\}$. For continuous treatments, it brings a high computational cost to consider all the discrete treatments received by the samples. To alleviate this, we adopt some sampled values evenly distributed in $\mathcal{T}$ to construct the set $\widehat{\mathcal{T}}$. It is worth mentioning that for each $t \in \widehat{\mathcal{T}}$, all the samples are assigned by the weights $\hat{q}_t(x)$ and taken into consideration for distribution alignment, avoiding the issue of data splitting. Finally, we achieve the following optimization problem

$$
\min_{\phi, \psi, \theta} \; \widehat{\mathcal{L}} + \lambda \sum_{t \in \widehat{\mathcal{T}}} \mathcal{W}(c_\phi, \hat{q}_t, \hat{q}),
\tag{20}
$$

where $\lambda$ is the trade-off hyper-parameters between the outcome prediction loss and the distribution discrepancies, $\phi$, $\psi$, and $\theta$ are implemented by neural networks. Figure 1 illustrate the framework of our propose method ORIC, and Algorithm 1 summarizes the major procedure of ORIC.

---

**Algorithm 1 O**ptimal transport for **R**educing b**I**as in **C**ontinuous treatment (ORIC).

---

**Input:** Training samples $\{x_i, t_i, y_i\}_{i=1}^{n}$.
**Initialize:** Representation learning model $\phi$, potential outcome prediction model $\psi$, generalized propensity score estimator $\theta$.
1: **repeat**
2:     Calculate the cost $c_\phi(x_i, x_j) = \|\phi(x_i) - \phi(x_j)\|_2$.
3:     **for all** $t \in \widehat{\mathcal{T}}$ **do**
4:         Calculate the outcome prediction loss according to Eq. (17).
5:         Estimate $\hat{q}_t(x_i)$ based on the normalized generalized propensity scores $\theta(\phi(x_i))$.
6:         Obtain the optimal transport plans $\tilde{\pi}^t$ by solving Problem (19).
7:         Calculate the Wasserstein discrepancies based on $\tilde{\pi}^t$ according to Eq. (18).
8:     **end for**
9:     Update $\phi$, $\psi$, and $\theta$ based on the gradient of Eq. (20).
10: **until** Convergence.

---

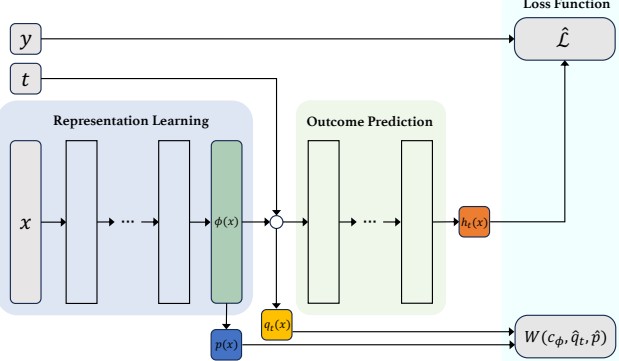

Figure 1: Overview of our proposed method ORIC.

## 5 EXPERIMENTS

In this section, we present experimental settings and results of continuous and binary treatments. The detailed experiments are provided in Appendix D and E.

### 5.1 CONTINUOUS TREATMENTS

**Dataset.** For the experiments of continuous treatments, we evaluate the performance of the proposed method using one synthetic dataset and two semi-synthetic datasets: IHDP (Hill, 2011) and News (Newman, 2008). The synthetic dataset consists of 500 training samples and 200 testing samples, with the parameter $\beta$ adjusted to simulate various confounding biases. IHDP contains 747 subjects, with 25 covariates for each sample to capture the aspects of children and their mothers. News contains 3,000 news items randomly sampled from Newman (2008), which simulates the opinions of a media consumer when exposed to multiple news items. We follow a similar approach in Nie et al. (2021) to generate continuous treatments and outcomes, and randomly divide the samples into a training set (67%) and a testing set (33%). The detailed synthesis protocols can be found in Appendix D.

**Compared methods.** We conduct comparison of our ORIC model with several compared methods, including the traditional statistical and machine learning method BART (Chipman et al., 2010), KNN (Peterson, 2009), GPS (Imbens, 2000), and modern neural network based methods MLP, DR-Net (Schwab et al., 2020), ADMIT (Wang et al., 2022), ACFR (Kazemi & Ester, 2024), and VCNet (Nie et al., 2021). Specifically, for GPS, in order to enhance the traditional statistical learning approach, we incorporate a Multilayer Perceptron Network for optimization (GPS+MLP). For VCNet, we consider the naive version of VCNet (VCNet) and VCNet with the target regularization (VCNet+TR).

**Evaluation Metrics.** Following Nie et al. (2021), we adopt the **A**verage **D**ose-**R**esponse **F**unction (ADRF) curve and $\sqrt{AMSE}$ as metrics. ADRF curve is the expected potential outcome under the treatment value $t$, which is defined as $\mu_t = \mathbb{E}[Y(t)]$. And $AMSE$ is defined in Eq. (4). We repeatedly carry out 100 trials on the simulated and the IHDP datasets, 20 trials on the News dataset, and report the mean and standard deviation of the results on the test set.

**Results and Discussions.** Table 1 presents the results of ORIC and the compared algorithms. Overall, the results indicate that ORIC consistently outperforms other methods on both synthetic and semi-synthetic datasets, showing the effectiveness of the proposed method. Typically, compared with traditional statistical method (*i.e.*, KNN, BART, GPS), neural network-based methods usually achieve performance improvement across a variety of datasets. Among the neural network methods, we observe that VCNet+TR outperforms other methods, showing the advantage the doubly robust property obtained by the targeted regularization. However, it lacks an explicit mechanism of distribution alignment to address confounding bias. ADMIT and DRNet split training samples into multiple smaller groups for training, suffering from the issue of data reduction for distribution alignment. Compared with them, ORIC involves all the training samples without data splitting for distribution alignment, and reduces the confounding bias and the outcome estimation error jointly, achieving the best performance in different kinds of datasets. In addition, ORIC obtains promising performance with different values of $\beta$, which demonstrates the robustness of the proposed method for different levels of confounding bias. Furthermore, from the ADRF curve in Figure 2, we observe that compared to VCNet, which achieves the best $\sqrt{AMSE}$ performance among other models, ORIC exhibits a significant improvement in fitting from left to right across synthetic($\beta = 0.25$), IHDP, and News datasets.

| Methods | Synthetic | | | | IHDP | News |
|---|---|---|---|---|---|---|
| | $\beta = 0.25$ | $\beta = 0.5$ | $\beta = 0.75$ | $\beta = 1$ | | |
| KNN | $0.2339 \pm 0.0294$ | $0.2234 \pm 0.0296$ | $0.2211 \pm 0.0235$ | $0.2361 \pm 0.0209$ | $0.8364 \pm 0.0917$ | $0.6104 \pm 0.4117$ |
| BART | $0.2205 \pm 0.0248$ | $0.2108 \pm 0.0312$ | $0.2177 \pm 0.0259$ | $0.2238 \pm 0.0212$ | $0.6825 \pm 0.0715$ | $0.5639 \pm 0.3125$ |
| GPS | $0.2103 \pm 0.0319$ | $0.2056 \pm 0.0345$ | $0.2063 \pm 0.0264$ | $0.2219 \pm 0.0238$ | $0.7247 \pm 0.0582$ | $0.4422 \pm 0.2033$ |
| MLP | $0.2083 \pm 0.0275$ | $0.2042 \pm 0.0311$ | $0.2044 \pm 0.0252$ | $0.2185 \pm 0.0202$ | $0.6566 \pm 0.0710$ | $0.4355 \pm 0.2098$ |
| MLP+GPS | $0.2077 \pm 0.0238$ | $0.2028 \pm 0.0203$ | $0.2022 \pm 0.0210$ | $0.2161 \pm 0.0157$ | $0.6303 \pm 0.0826$ | $0.4255 \pm 0.2115$ |
| DRNet | $0.1992 \pm 0.0303$ | $0.2033 \pm 0.0226$ | $0.1967 \pm 0.0172$ | $0.2046 \pm 0.0195$ | $0.5714 \pm 0.0211$ | $0.2380 \pm 0.0141$ |
| ADMIT | $0.1542 \pm 0.0325$ | $0.1729 \pm 0.0467$ | $0.1856 \pm 0.0345$ | $0.1645 \pm 0.0279$ | $0.5222 \pm 0.0375$ | $0.1832 \pm 0.0394$ |
| ACFR | $0.1428 \pm 0.0259$ | $0.1651 \pm 0.0325$ | $0.1654 \pm 0.0334$ | $0.1567 \pm 0.0248$ | $0.5134 \pm 0.0523$ | $0.1719 \pm 0.0767$ |
| VCNet | $0.1233 \pm 0.0328$ | $0.1577 \pm 0.0460$ | $0.1543 \pm 0.0536$ | $0.1395 \pm 0.0369$ | $0.4656 \pm 0.0476$ | $0.1905 \pm 0.1072$ |
| VCNet+TR | $0.1155 \pm 0.0305$ | $0.1361 \pm 0.0439$ | $0.1442 \pm 0.0512$ | $0.1257 \pm 0.0381$ | $0.3712 \pm 0.0465$ | $0.1675 \pm 0.0566$ |
| ORIC | $\mathbf{0.1098 \pm 0.0273}$ | $\mathbf{0.1234 \pm 0.0388}$ | $\mathbf{0.1313 \pm 0.0464}$ | $\mathbf{0.1168 \pm 0.0316}$ | $\mathbf{0.3595 \pm 0.0304}$ | $\mathbf{0.1507 \pm 0.0406}$ |

Table 1: Comparison of ORIC with baseline algorithms of related networks. The $\pm$ denotes the mean and standard deviation of $\sqrt{AMSE}$.

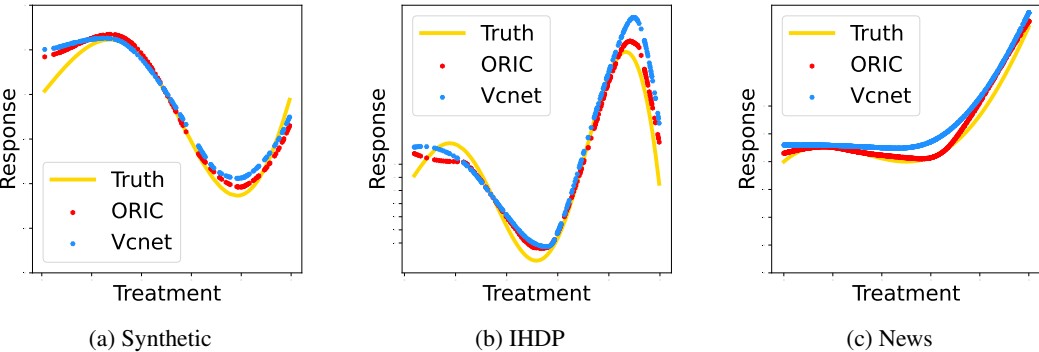

(a) Synthetic      (b) IHDP      (c) News

Figure 2: Presented from left to right are the ADRF results for the Synthetic, IHDP, and News datasets. The yellow line illustrates the true results, while the blue points represent the estimates synthesized by VCNet, and the red points correspond to the estimates produced by ORIC.

## 5.2 BINARY TREATMENTS

**Dataset.** We conduct experiments on two semi-synthetic datasets, IHDP (Brooks-Gunn et al., 1992) and News (Newman, 2008). For the IHDP dataset, we randomly select 100 datasets from the IHDP-1000 version and follow (Shalit et al., 2017) to split training and testing sets. In the News dataset, we assign the first 3,500 samples to the training set and 1,000 samples as the test set (Johansson et al., 2016). Furthermore, experiments on synthetic data are provided in Appendix E.

**Compared methods.** We evaluate the proposed method in the binary treatment setting with several baselines, including non-neural network methods BART, kNN, OLS, and neural network methods MLP, CFR (Shalit et al., 2017), GANITE (Yoon et al., 2018), Dragonnet (Shi et al., 2019), DKLite (Zhang et al., 2020), CausalOT (Li et al., 2021), ESCFR (Wang et al., 2024).

**Evaluation Metrics.** For the synthetic dataset, we adopt mean absolute errors(MAE) (Dehejia & Wahba, 1999) as metric, which is defined as $MAE = |\widehat{ATE} - ATE|$ between predicted average treatment effect and ground truth. For semi-synthetic datasets, besides $MAE$, we adopt $\sqrt{PEHE}$ (Hill, 2011) and $\sqrt{AMSE}$ to evaluate the proposed method. **P**recision in **E**stimation of **H**eterogeneous **E**ffect (PEHE) is defined as $\sqrt{PEHE} = \sqrt{\frac{1}{n}\sum_{i=1}^{n}[(h_1(x_i) - h_0(x_i)) - (\mu_1(x_i) - \mu_0(x_i))]^2}$. The definition of AMSE is the same as Eq.(4), with $\mathcal{T} \in \{0, 1\}$.

**Results and Discussions.** Tables 2 demonstrate the result across two semi-synthetic datasets in the binary setting. We draw similar observations from the results of the binary treatment setting to the continuous treatment setting. Benefit from the mechanism that involves all the samples for training to avoid data splitting, ORIC achives the best or highly competitive performance compared with other methods. This observation demonstrates that ORIC not only can handle continuous treatment, but also obtain promising performance in binary treatment, indicating the capability of generalization in different kinds of treatment settings.

| Methods | IHDP | | | News | | |
| | $\sqrt{PEHE}$ | $MAE$ | $\sqrt{AMSE}$ | $\sqrt{PEHE}$ | $MAE$ | $\sqrt{AMSE}$ |
| --- | --- | --- | --- | --- | --- | --- |
| BART | $13.8853 \pm 9.3630$ | $9.1204 \pm 3.0154$ | $10.0374 \pm 7.2281$ | $7.3663 \pm 2.2189$ | $5.6858 \pm 1.7925$ | $5.6355 \pm 1.6655$ |
| OLS | $14.3736 \pm 11.3114$ | $8.8191 \pm 2.5947$ | $9.7246 \pm 6.9604$ | $8.0871 \pm 2.3580$ | $5.7820 \pm 1.6172$ | $6.3790 \pm 1.8565$ |
| MLP | $15.3081 \pm 11.2789$ | $8.9105 \pm 3.1171$ | $11.0619 \pm 8.5434$ | $8.2535 \pm 2.4681$ | $5.3473 \pm 1.6470$ | $6.0092 \pm 1.7761$ |
| KNN | $3.1108 \pm 3.8114$ | $0.4104 \pm 0.6477$ | $9.7638 \pm 7.4574$ | $7.0048 \pm 2.3408$ | $5.1976 \pm 2.0301$ | $5.5409 \pm 1.7343$ |
| CFRNet | $1.2809 \pm 1.7304$ | $\mathbf{0.1582 \pm 0.1986}$ | $1.2739 \pm 1.7038$ | $2.0527 \pm 0.6464$ | $0.3080 \pm 0.2224$ | $2.4187 \pm 0.6538$ |
| Dragonnet | $1.4305 \pm 1.8883$ | $0.2672 \pm 0.4576$ | $1.3229 \pm 1.7893$ | $1.7916 \pm 0.5652$ | $0.3531 \pm 0.1724$ | $3.8169 \pm 1.6722$ |
| GANITE | $5.0500 \pm 1.3205$ | $4.2490 \pm 0.6251$ | $13.4438 \pm 6.7216$ | $2.6473 \pm 0.6873$ | $2.6375 \pm 0.6867$ | $6.1070 \pm 1.1409$ |
| DKLite | $5.3315 \pm 7.0602$ | $0.5472 \pm 0.7026$ | $5.7984 \pm 7.1115$ | $1.8172 \pm 0.5182$ | $0.2328 \pm 0.1272$ | $\mathbf{1.9610 \pm 0.5701}$ |
| ESCFR | $1.2443 \pm 2.1300$ | $0.4112 \pm 0.5902$ | $1.3498 \pm 2.1298$ | $2.7671 \pm 0.8924$ | $0.8651 \pm 0.6514$ | $2.9547 \pm 0.8822$ |
| CausalOT | $13.8269 \pm 13.5417$ | $2.4498 \pm 0.8065$ | $7.3281 \pm 6.2416$ | $9.1213 \pm 2.0943$ | $2.3308 \pm 0.4832$ | $4.1533 \pm 1.0084$ |
| ORIC | $\mathbf{1.1129 \pm 1.4290}$ | $0.2134 \pm 0.3488$ | $\mathbf{1.1976 \pm 1.3822}$ | $\mathbf{1.7183 \pm 0.5488}$ | $\mathbf{0.1624 \pm 0.1587}$ | $2.3972 \pm 0.5678$ |

Table 2: Comparison of ORIC with baseline algorithms of related neural-network and non-neural-network on semi-synthetic dataset. Specifically, we perform over 100 trials on the IHDP dataset, and 50 trials on the News dataset.

## 6 CONCLUSION

In this paper, we estimate the effect of binary and continuous treatments by reducing the confounding bias from non-RCTs. We characterize the confounding bias by different probability measures of the same set of all the samples, and analyze the confounding bias and outcome prediction error based on optimal transport built on probability measures. Motivated by this, we propose to learn balanced representations to reduce the outcome estimation error and the confounding bias simultaneously. By doing this, we avoid data reduction from splitting which is commonly used in existing methods, and enhance the generalization ability of the model. We conduct experiments on both binary and continuous settings and synthetic and semi-synthetic datasets are adopted. The experimental results demonstrate the effectiveness of the proposed method.

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

## A  IMPLEMENTATION OF $\theta$

The implementation of $\theta$ is based on (HiranoK, 2004) and described as follows. Assuming that the conditional distribution of treatment given covariates is Gaussian, i.e., $P(t \mid x_i) \sim \mathcal{N}(\theta(\phi(x_i)), \sigma^2)$. We can estimate the parameters by maximizing the likelihood:

$$\max_{\theta, \sigma} L(\hat{\theta}, \hat{\sigma}; t, x) := \prod_{i=1}^{n} \frac{1}{\sqrt{2\pi\sigma^2}} \exp\left(-\frac{1}{2\sigma^2}(t_i - \theta(\phi(x_i)))^2\right). \tag{21}$$

After that, the estimated generalized propensity score is given by:

$$\hat{p}(t \mid x_i) = \frac{1}{\sqrt{2\pi\hat{\sigma}^2}} \exp\left(-\frac{1}{2\hat{\sigma}^2}(t - \hat{\theta}(\phi(x_i)))^2\right). \tag{22}$$

## B  THEORETICAL ANALYSIS REGARDING PEHE

We use binary treatment as an example to illustrate that our theoretical results can be applied to PEHE. Based on the assumptions in Theorem 2, we first decompose PEHE for the true causal effect $\tau(x) = \mu_1(x) - \mu_0(x)$ as follows:

$$\begin{aligned}
\varepsilon_{PEHE} &= E_{x \sim q(x)}[\ell(h_1(x) - h_0(x), \mu_1(x) - \mu_0(x))] \\
&\leq E_{x \sim q(x)}[\ell(h_1(x), \mu_1(x))] + E_{x \sim q(x)}[\ell(h_0(x), \mu_0(x))] \\
&= \varepsilon_q(h_1) + \varepsilon_q(h_0)
\end{aligned} \tag{23}$$

where $\ell$ is the $L_p$-norm based loss function and has the triangle inequality property.

And we define the estimation error of the potential outcome function $\mu_1(x)$ and $\mu_0(x)$ in treatment and control groups, respectively:

$$\varepsilon_{q_1}(h_1) = E_{x \sim q_1(x)}\ell(h_1(x), \mu_1(x)) \tag{24}$$

$$\varepsilon_{q_0}(h_0) = E_{x \sim q_0(x)}\ell(h_0(x), \mu_0(x)) \tag{25}$$

According to Eq. 12, we have

$$\varepsilon_{PEHE} \leq \varepsilon_q(h_1) + \varepsilon_q(h_0) \leq \varepsilon_{q_1}(h_1) + \varepsilon_{q_0}(h_0) + \mathcal{W}(c, q_1, q) + \mathcal{W}(c, q_0, q) \tag{26}$$

## C  PROOFS OF THEOREMS

### C.1  PROOF OF THEOREM 1

According to the definition of $\xi(m, t)$, we have:

$$\begin{aligned}
\xi(m, t) &= \|\mathbb{E}_{x \sim q_t(x)}m(x) - \mathbb{E}_{x \sim q(x)}m(x)\| \\
&= \left\| \int_{\mathcal{X}} m(x)dq_t(x) - \int_{\mathcal{X}} m(x)dq(x) \right\| \tag{27} \\
&\leq \sup_{m(x)-m(x')\leq c(x,x')} \int_{\mathcal{X}} m(x)dq_t(x) - \int_{\mathcal{X}} m(x)dq(x) \tag{28} \\
&\leq \inf_{\pi \in \Pi(q_t, q)} \int_{\mathcal{X} \times \mathcal{X}} c(x, x')d\pi(x, x') \tag{29} \\
&= \mathcal{W}(c, q_t, q). \tag{30}
\end{aligned}$$

Under the assumption of Theorem 1, Eq. (28) is the the worst-case of Eq. (27), and Eq. (29) holds because of the property of the dual problem, which just corresponding to the definition of the Wasserstein distance. As a result, we obtain $\xi(m, t) \leq \mathcal{W}(c, q_t, q)$, which finishes the proof by integrating $p(t)$ on both sides of the inequality.

## C.2    PROOF OF THEOREM 2

According to (Saitoh, 2020), $\ell(h_t(x), \mu_t(x))$ also belongs to the RKHS $\mathcal{H}$ since it is a convex loss-function defined on $h_t, \mu_t \in \mathcal{F}$. As a result, $\ell$ has the reproducing property and the norn $\|\ell\|$ is bounded. For simplicity, we assume that $\|\ell\|$ is bounded by 1, which is easily extendable to the case when $\|\ell\| \leq M$ by scaling (Redko et al., 2017). Now, the estimation error can be expressed in terms of the inner product in the corresponding Hilbert space ,

$$\varepsilon_q(h_t) = \mathbb{E}_{x \sim q(x)} \ell(h_t(x), \mu_t(x)) = \mathbb{E}_{x \sim q(x)}[\langle \phi(x), \ell \rangle_{\mathcal{H}}], \tag{31}$$

$$\varepsilon_{q_t}(h_t) = \mathbb{E}_{x \sim q_t(x)} \ell(h_t(x), \mu_t(x)) = \mathbb{E}_{x \sim q_t(x)}[\langle \phi(x), \ell \rangle_{\mathcal{H}}]. \tag{32}$$

With $\varepsilon_q(h_t) = \varepsilon_q(h_t) + \varepsilon_{q_t}(h_t) - \varepsilon_{q_t}(h_t)$ and the above definitions, we have :

$$\begin{aligned}
\varepsilon_q(h_t) - \varepsilon_{q_t}(h_t) &= \mathbb{E}_{x' \sim q(x)}[\langle \phi(x'), \ell \rangle_{\mathcal{H}}] - \mathbb{E}_{x \sim q_t(x)}[\langle \phi(x), \ell \rangle_{\mathcal{H}}] \\
&= \langle \mathbb{E}_{x' \sim q(x)}[\phi(x')] - \mathbb{E}_{x \sim q_t(x)}[\phi(x)], \ell \rangle_{\mathcal{H}} \\
&\leq \|\ell\|_{\mathcal{H}} \|\mathbb{E}_{x' \sim q(x)}[\phi(x')] - \mathbb{E}_{x \sim q_t(x)}[\phi(x)]\|_{\mathcal{H}} \\
&\leq \| \int_{\mathcal{X}} \phi d(q_t(x) - q(x))\|_{\mathcal{H}}.
\end{aligned} \tag{33}$$

The first line is obtained by the reproducing property of $\ell$, and the last line is due to $\|\ell\| \leq 1$. Now using the definition of the joint distribution we have:

$$\begin{aligned}
\| \int_{\mathcal{X}} \phi d(q_t(x) - q(x))\|_{\mathcal{H}} &= \| \int_{\mathcal{X} \times \mathcal{X}} (\phi(x) - \phi(x')) d\pi(x, x')\|_{\mathcal{H}} \\
&\leq \int_{\mathcal{X} \times \mathcal{X}} \|\phi(x) - \phi(x')\|_{\mathcal{H}} d\pi(x, x') \\
&\leq \inf_{\pi \in \Pi(q_t, q)} \int_{\mathcal{X} \times \mathcal{X}} \|\phi(x) - \phi(x')\|_{\mathcal{H}} d\pi(x, x') \tag{34} \\
&= \mathcal{W}(c, q_t, q), \tag{35}
\end{aligned}$$

where $x \sim q_t(x)$ and $x' \sim q(x)$. As a result, we get $\varepsilon_p(h_t) - \varepsilon_{q_t}(h_t) \leq \mathcal{W}(c, q_t, q)$, which finishes the proof by integrating $p(t)$ on both sides of the inequality.

## C.3    PROOF OF THEOREM 3

With the triangular inequality of the Wasserstein metric, we have:

$$\begin{aligned}
\mathcal{W}(c, q_t, q) &\leq \mathcal{W}(c, q_t, \hat{q}_t) + \mathcal{W}(c, \hat{q}_t, q) \\
&\leq \mathcal{W}(c, q_t, \hat{q}_t) + \mathcal{W}(c, \hat{q}_t, \hat{q}) + \mathcal{W}(c, \hat{p}, q) \\
&= \mathcal{W}(c, q_t, \hat{q}_t) + \mathcal{W}(c, q, \hat{q}) + \mathcal{W}(c, \hat{q}_t, \hat{p}) \tag{36}
\end{aligned}$$

Next, we present Lemma 1 showing the convergence of the empirical measure $\hat{\mu}$ to its true $\mu$ w.r.t. the Wasserstein metric, which allows us to propose a generalization bound based on the Wasserstein distance for finite samples rather than true population measures:

**Lemma 1** *((Bolley et al., 2007), Theorem 1.1). Let $\mu$ be a probability measure in $\mathbb{R}^d$ satisfying $T_1(zeta)$ inequality, and $\hat{\mu} = \frac{1}{n} \sum_{i=1}^{n} \delta_{x_i}$ be its associated empirical measure with $n$ units. Then for any $d' > d$ and $\zeta' < \zeta$, there exists some constant $n_0$ depending on $d'$ and some square exponential moment of $\mu$ such that for any $\epsilon > 0$ and $n \geq n_0 \max(\epsilon^{-(d'+2)}, 1)$*

$$\mathbb{P}[W_1(\mu, \hat{\mu}) > \epsilon] \leq \exp\left(-\frac{\zeta' n \epsilon^2}{2}\right), \tag{37}$$

*where $d', \zeta'$ can be calculated explicitly.*

The Hoeffding inequality in Lemma 1 gives the following inequality which holds with the probability at least $1 - \delta$:

$$\mathcal{W}(c, q_t, \hat{q}_t) \leq \sqrt{2 \log\left(\frac{1}{\delta}\right) / \zeta' n}, \ \ \mathcal{W}(c, \hat{q}, q) \leq \sqrt{2 \log\left(\frac{1}{\delta}\right) / \zeta' n}. \tag{38}$$

Combining Eq. (36) and Eq. (38) together, we have:

$$W(c, q_t, p) \leqslant \sqrt{2 \log \left(\frac{1}{\delta}\right) / \zeta' n} + \sqrt{2 \log \left(\frac{1}{\delta}\right) / \zeta' n} + W(c, \hat{q}_t, \hat{p})$$

$$= W(c, \hat{q}_t, \hat{p}) + 2\sqrt{2 \log \left(\frac{1}{\delta}\right) / \zeta' n}$$

$$:= W(c, \hat{q}_t, \hat{p}) + \mathcal{O}\left(1/\sqrt{\delta n}\right), \tag{39}$$

which finishes the proof.

## D EXPERIMENTS OF CONTINUOUS TREATMENTS

### D.1 EXPERIMENTAL SETTINGS

**Synthetic.** We synthesize data as follows: $x_j \overset{\text{i.i.d.}}{\sim} \text{Unif } [0,1]$, where $x_j$ is the $j$-th dimension of $x \in \mathcal{R}^6$, and generate treatment and outcome as:

$$\tilde{t} \mid x = \frac{10 \sin \left(\max \left(x_1, x_2, x_3\right)\right) + \max \left(x_3, x_4, x_5\right)^3}{1 + (x_1 + x_5)^2} + \sin \left(\beta x_3\right) \left(1 + \exp \left(x_4 - \beta x_3\right)\right)$$

$$+ x_3^2 + 2 \sin \left(x_4\right) + 2 x_5 - 6.5 + \mathcal{N}(0, 0.25)$$

$$y \mid x, t = \cos(2\pi(t - \beta)) \left(t^2 + \frac{4 \max \left(x_1, x_6\right)^3}{1 + 2 x_3^2} \sin \left(x_4\right)\right) + \mathcal{N}(0, 0.25)$$

where $t = (1 + \exp(-\tilde{t}))^{-1}, \beta = \{0.25, 0.5, 0.75, 1\}$. It is noteworthy that $\pi(t \mid x)$ only is contingent upon $x_1, x_2, x_3, x_4, x_5$ while $Q(t, x)$ only is contingent upon $x_1, x_3, x_4, x_6$.

**IHDP.** The original semi-synthetic IHDP dataset from Hill (2011) includes binary treatments, comprising 747 observations across 25 covariates. To facilitate comparisons using continuous treatments, we randomly synthesize both treatment and response variables as follows:

$$\tilde{t} \mid x = \frac{2 x_1}{(1 + x_2)} + \frac{2 \max \left(x_3, x_5, x_6\right)}{2 + \min \left(x_3, x_5, x_6\right)} + 2 \tanh \left(5 \frac{\sum_{i \in S_{dis,2}} \left(x_i - c_2\right)}{|S_{dis,2}|}\right) - 4 + \mathcal{N}(0, 0.25)$$

$$y \mid x, t = \frac{\sin(3\pi t)}{1.2 - t} \left(\tanh \left(5 \frac{\sum_{i \in S_{dis,1}} \left(x_i - c_1\right)}{|S_{dis,1}|}\right) + \frac{\exp \left(2 \left(x_1 - x_6\right)\right)}{0.5 + 5 \min \left(x_2, x_3, x_5\right)}\right) + \mathcal{N}(0, 0.25)$$

where $t = (1 + \exp(-\tilde{t}))^{-1}, S_{\text{con}} = \{1, 2, 3, 5, 6\}$ is the index set of continuous features, $S_{\text{dis},1} = \{4, 7, 8, 9, 10, 11, 12, 13, 14, 15\}, S_{\text{dis},2} = \{16, 17, 18, 19, 20, 21, 22, 23, 24, 25\}$ and $S_{\text{dis},1} \cup S_{\text{dis},2} = [25] - S_{\text{con}}$. Here $c_1 = \mathcal{E} \frac{\sum_{i \in S_{\text{dis},1}} x_i}{|S_{\text{dis},1}|}, c_2 = \mathcal{E} \frac{\sum_{i \in S_{\text{dis},2}} x_i}{|S_{\text{dis},2}|}$. It is noteworthy that all continuous features are advantageous for $\pi(t \mid x)$ and $Q(t, x)$ but only $S_{\text{dis},1}$ is advantageous for $Q$ and only $S_{\text{dis},2}$ is advantageous for $\pi$. Following Hill (2011), covariates are standardized to have a mean of 0 and a standard deviation of 1, while the synthesized treatment values are normalized to the range $[0, 1]$. Furthermore, we applied denoising techniques to the error data produced during the construction of the IHDP dataset.

**News.** The News dataset comprises 3,000 randomly sampled news items from the NY Times corpus (Newman, 2008), originally introduced as a benchmark for binary treatment settings (Johansson et al., 2016). We synthesize the treatment and outcome variables similarly to the method outlined in Bica et al. (2020). We first synthesize $v_1', v_2'$ and $v_3'$ from $\mathcal{N}(0, 1)$ and then set $v_i = v_i' / \|v_i'\|_2$ for $i = \{1, 2, 3\}$. Given $x$, we synthesize $t$ from $\text{Beta} \left(2, \left|\frac{v_3^\top x}{2 v_2^\top x}\right|\right)$. And we synthesize the outcome by

$$y' \mid x, t = \exp\left(\frac{v_2^\top x}{v_3^\top x} - 0.3\right)$$

$$y \mid x, t = 2\left(\max\left(-2, \min\left(2, y'\right)\right) + 20 v_1^\top x\right) * \left(4(t - 0.5)^2 * \sin\left(\frac{\pi}{2} t\right)\right) + \mathcal{N}(0, 0.5)$$

## D.2 SENSITIVITY ANALYSIS

To empirically study the effect of the hyper-parameter $\lambda$ in Eq. (20) which trades off between the outcome prediction loss and the Wasserstein discrepancies, we conduct experiments on synthetic dataset($\beta = 0.25$) with varying values of $\lambda$ in the range $[0.5, 1.3]$, and present the results of $\sqrt{AMSE}$ in Figure 3(a). We observe that ORIC is able to achieve good performance with a wide range of the values of $\lambda$, which verifies the sensitivity of ORIC with respect to $\lambda$. Besides, we conduct experiments on synthetic dataset with different numbers of sampled treatment values in the discrete set $\widehat{\mathcal{T}}$, and report the results of $\sqrt{AMSE}$ in Figure 3(b). We observe that ORIC stably achieves promising performance when the number of discrete values of the treatment is greater than 50, since more values of the treatment provide finer-grained estimation for the conditional marginal distribution $\hat{q}_t(x)$.

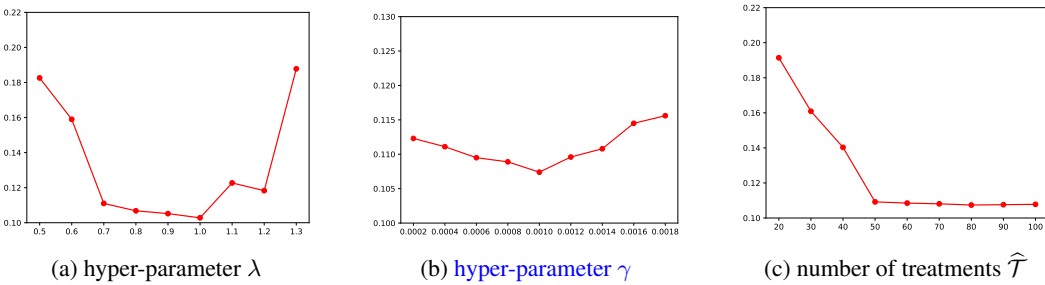

|                           |                              |                                      |
| (a) hyper-parameter $\lambda$ | (b) hyper-parameter $\gamma$ | (c) number of treatments $\widehat{\mathcal{T}}$ |

Figure 3: Figure (a) demonstrates the trade-off of the hyperparameter $\lambda$ between the outcome prediction loss and the Wasserstein discrepancies with the variation of $\lambda$ values range from $[0.5, 1.3]$, and present the results of $\sqrt{AMSE}$. Figure (b) demonstrates the trade-off of the entropy regularization hyperparameter $\gamma$ values range from $[0.0001, 0.1]$, and present the results of $\sqrt{AMSE}$. Figure (c) illustrates the experiment on synthetic dataset with different numbers of sampled treatment values in the discrete set $\widehat{\mathcal{T}}$, and report the results of $\sqrt{AMSE}$.

## E EXPERIMENTS OF BINARY TREATMENTS

### E.1 EXPERIMENT SETTINGS

**Synthetic.** Following the similar protocols in (Yao et al., 2018; Hatt & Feuerriegel, 2021), We generate a synthetic dataset in binary treatment setting as follow:

We employ a Gaussian mixture model consisting of two distributions: $\mathcal{N}_1 = \mathcal{N}\left(0.5^{10 \times 1}, 0.5 \times \Sigma_1 \Sigma_1^T\right), \mathcal{N}_2 = \mathcal{N}\left(1^{10 \times 1}, 0.5 \times \Sigma_2 \Sigma_2^T\right)$, where $\Sigma_1 \sim \mathcal{U}\left((0, 0.5)^{10 \times 10}\right), \Sigma_2 \sim \mathcal{U}\left((0, 1)^{10 \times 10}\right)$. We then synthesize 1,500 treated and control samples from $\mathbf{x}^t \sim \alpha_t \mathcal{N}_1 + (1 - \alpha_t) \mathcal{N}_2, x^c \sim \alpha_c \mathcal{N}_1 + (1 - \alpha_c) \mathcal{N}_2$, fix $\alpha_t$ to 0.5 and vary the value of $\alpha_c$ to simulate different confounding bias. The outcomes are defined as $y = \sin\left(w_1^\top x\right) + \cos\left(w_2^\top (x \odot x)\right) + t + \epsilon$, where $w. \sim \mathcal{U}\left((0, 1)^{10 \times 1}\right), \epsilon \sim \mathcal{N}(0, 0.1)$.

### E.2 RESULTS AND DISCUSSIONS

#### E.2.1 CONTINUOUS TREATMENT SETTING

Table 3 illustrates ORIC ablation study on the loss function involving Wasserstein distances.

| Methods | Synthetic | | | | IHDP | News |
| --- | --- | --- | --- | --- | --- | --- |
| | $\beta = 0.25$ | $\beta = 0.5$ | $\beta = 0.75$ | $\beta = 1$ | | |
| ORIC without wass | $0.2083 \pm 0.0275$ | $0.2042 \pm 0.0311$ | $0.2044 \pm 0.0252$ | $0.2185 \pm 0.0202$ | $0.6566 \pm 0.0710$ | $0.4355 \pm 0.2098$ |
| ORIC without wass and gps | $0.2077 \pm 0.0238$ | $0.2028 \pm 0.0203$ | $0.2022 \pm 0.0210$ | $0.2161 \pm 0.0157$ | $0.6303 \pm 0.0826$ | $0.4255 \pm 0.2115$ |
| ORIC | $\mathbf{0.1098 \pm 0.0273}$ | $\mathbf{0.1234 \pm 0.0388}$ | $\mathbf{0.1313 \pm 0.0464}$ | $\mathbf{0.1168 \pm 0.0316}$ | $\mathbf{0.3595 \pm 0.0304}$ | $\mathbf{0.1507 \pm 0.0406}$ |

Table 3: Ablation study on the loss function involving Wasserstein distances. The $\pm$ denotes the mean and standard deviation of $\sqrt{AMSE}$.

Table 4 presents the computational time for one realization of ORIC on the synthetic($\beta = 0.25$) dataset.

| Methods | ORIC | VCNet+TR | VCNet | ADMIT | ACFR | DRNet | GPS+MLP | MLP | GPS | BART | KNN |
| --- | --- | --- | --- | --- | --- | --- | --- | --- | --- | --- | --- |
| **Times** | 135s | 23s | 17s | 47s | 24s | 26s | 25s | 18s | 9s | 7s | 8s |

Table 4: Execution time results on synthetic($\beta = 0.25$) dataset.

### E.2.2 BINARY TREATMENT SETTING

Tables 5 illustrate the result of synthetic data in different bias situation, which has a similar observation as in continuous setting. ORIC outperforms other methods and achieve the best result in different levels of confounding bias, indicating the superior performance of robustness.

| Methods | Synthetic | | | |
| --- | --- | --- | --- | --- |
| | $\alpha_c = 0.2$ | $\alpha_c = 0.4$ | $\alpha_c = 0.6$ | $\alpha_c = 0.8$ |
| BART | $0.0622 \pm 0.0374$ | $0.0484 \pm 0.0194$ | $0.0255 \pm 0.0206$ | $0.0397 \pm 0.0207$ |
| OLS | $0.0568 \pm 0.0420$ | $0.0471 \pm 0.0361$ | $0.0387 \pm 0.0234$ | $0.0412 \pm 0.0259$ |
| MLP | $0.0862 \pm 0.0813$ | $0.0803 \pm 0.0600$ | $0.4992 \pm 0.0422$ | $0.0621 \pm 0.0388$ |
| KNN | $0.0229 \pm 0.0196$ | $0.0276 \pm 0.0198$ | $0.0306 \pm 0.0184$ | $0.0296 \pm 0.0259$ |
| CFRNet | $0.0328 \pm 0.0063$ | $0.0326 \pm 0.0065$ | $0.0383 \pm 0.0326$ | $0.0475 \pm 0.0345$ |
| Dragonnet | $0.0351 \pm 0.0104$ | $0.0323 \pm 0.0092$ | $0.04778 \pm 0.0061$ | $0.0482 \pm 0.0067$ |
| GANITE | $0.1883 \pm 0.0530$ | $0.1779 \pm 0.0672$ | $0.3219 \pm 0.0574$ | $0.3916 \pm 0.0581$ |
| DKLite | $0.0599 \pm 0.0338$ | $0.0432 \pm 0.0158$ | $0.0302 \pm 0.0344$ | $0.0753 \pm 0.0463$ |
| ORIC | $\mathbf{0.0052 \pm 0.0089}$ | $\mathbf{0.0282 \pm 0.0048}$ | $\mathbf{0.0235 \pm 0.0166}$ | $\mathbf{0.0291 \pm 0.0186}$ |

Table 5: Comparison of ORIC with baseline algorithms of related neural-network and non-neural-network on synthetic dataset. Specifically, we conducted over 10 trials on a synthetic dataset, adopting $MAE$ as the evaluation metric.

Table 6 presents the computational time for one realization of ORIC on the IHDP-1000 dataset.

| Methods | ORIC | CFRNet | DragonNet | DKLITE | ESCFR | CausalOT | GANITE | BART | OLS | KNN |
|---------|------|--------|-----------|--------|-------|----------|--------|------|-----|-----|
| **Times** | 76s | 47s | 41s | 4s | 165s | 4s | 4s | 0.2s | 0.2s | 0.3s |

Table 6: Execution time results on IHDP-1000 dataset.

