# OpenReview forum: "Optimal Transport for Reducing Bias in Causal Inference without Data Splitting"
_ICLR.cc/2025/Conference — Submitted to ICLR 2025_

### Official Review · Reviewer_E4XK · 2024-10-17

**Soundness:** 2
**Presentation:** 3
**Contribution:** 2
**Rating:** 6
**Confidence:** 3

**Summary:**

The paper proposes a novel method for addressing covariate shift in observational data when estimating treatment effects, based on optimal transport. The benefit of the method is that data does not need to be split into groups and that the method can easily be applied to both binary and continuous treatments. Extensive theoretical and empirical results are provided to validate the efficacy of the method.

**Strengths:**

- The method addresses a clear problem and provides a natural solution.
- The paper provides extensive theoretical and empirical results.
- Despite the complex topic, the paper is generally easy to follow.

**Weaknesses:**

1. I found it hard to judge the novelty and contribution of the work with respect to existing work:
  - There is earlier work on applying optimal transport for treatment effect estimation, which is not discussed [1, 2]. How does this work compare to your own? Is there any reason why these methods were not included as benchmarks?
  - Apart from the data splitting, how exactly is your method different from CFR_Wass in Shalit et al. (2017)? Understanding this would make your own contribution more clear. Did you use CFR's version with Wasserstein distance in your experiments?

2. The method introduces significant complexity and I would like to see more discussion of technical details:
  - Tuning of hyperparameters ($\lambda$, $\gamma$), as well as tuning in general, is not discussed. As this is a difficult problem in CATE estimation generally, I would like to see how the authors addressed this. Additionally, a sensitivity analysis of $\lambda$ is provided, but not for $\gamma$. Is there any reason why no such analysis is presented for $\gamma$?
  - The efficiency of the method is not discussed. Does the Sinkhorn algorithm make your training procedure much slower?

3. The conclusion summarizes the work, but does not include limitations of the proposed method or suggestions for future work.

___
[1] Wang, H., Fan, J., Chen, Z., Li, H., Liu, W., Liu, T., ... & Tang, R. (2024). Optimal transport for treatment effect estimation. Advances in Neural Information Processing Systems, 36.

[2] Li, Q., Wang, Z., Liu, S., Li, G., & Xu, G. (2021). Causal optimal transport for treatment effect estimation. IEEE transactions on neural networks and learning systems, 34(8), 4083-4095.

**Questions:**

- See weaknesses points 1, 2, and 3

- Calling the potential outcome $\tau$ instead of $\mu$ is a bit confusing, as $\tau$ is typically reserved for the treatment effect. I would suggest calling it $\mu$. Related to this, previous work is mostly focused on estimating the treatment effect, instead of the potential outcome; or PEHE instead of AMSE. Could your theoretical results be applied to PEHE as well?

- Any reason why evaluation for continuous treatments was not done with the Mean Integrated Squared Error (MISE)?

**Minor points:**
- Missing space in line
- Missing space in line 845

---

> ### Author Response · Authors · 2024-11-25
>
> Thank you for the valuable and insightful comments.
> We will revise the submission according to the
> comments. Our responses are as follows.
>
> **Q1**: Comparison with [1,2]
>
> **A1**:
>
> 1) Both [1] and [2] consider the binary treatment setting. Different from them, we propose a unified framework for both binary and continuous treatment settings.
>
> 2) [1] reduce the distribution shift between the treatment group and the control group, i.e., $\mathcal{W}(q_1, q_0)$. Different from it, we reduce the shift between each conditional distribution and the marginal distribution, i.e., $\sum_{t} \mathcal{W}(q_t, q)$.
>
> 3) [2] considers an optimal transport problem between the factual and counterfactual distributions, which is different from our optimal transport model. In addition, [2] relies on a linear potential outcome model. We employ the neural network to train a nonlinear outcome model.
>
> 4) We have conducted the methods in [1,2] as the compared methods in the revision. Our method achieves better performance. The results are as follows.
>
> | Methods  | IHDP$(\sqrt{PEHE})$   | IHDP$(MAE)$         | IHDP$(\sqrt{AMSE})$ |
> |----------|-----------------------|---------------------|---------------------|
> | ORIB     | 1.1129 $\pm$ 1.4290   | 0.2134 $\pm$ 0.3488 | 1.1976 $\pm$ 1.3822 |
> | ESCFR    | 1.2443 $\pm$ 2.1300   | 0.4112 $\pm$ 0.5902 | 1.3498 $\pm$ 2.1298 |
> | CausalOT | 13.8269 $\pm$ 13.5417 | 2.4498 $\pm$ 0.8065 | 7.3281 $\pm$ 6.2416 |
>
> **Q2**: Difference from CFR-Wass.
>
> **A2**:
>
> Apart from data splitting, there are two major difference between our method and CFR-Wass.
>
> 1) CFR-Wass reduces the Wasserstein distance between the treatment group and the control group, i.e., $\mathcal{W}(q_1, q_0)$. Different from it, we reduce the Wasserstein distance between each conditional distribution and the marginal distribution, i.e., $\sum_{t} \mathcal{W}(q_t, q)$.
>
> 2) CFR-Wass assigns equal weights to the samples in one group. Different from it, we introduce the generalized propensity score into the optimal tranpsort model as the sample weights.
>
> Yes, we conduct CFR with the Wasserstein distance as a compared method in Section 5.2.
>
> **Q3**: Parameter sensitivity.
>
> **A3**: The hyperparameter $\lambda$ between the outcome prediction loss and the Wasserstein discrepancies range from $[0.5, 1.3]$. The trade-off hyperparameter of the entropy regularization $\gamma$ is tuned in the range $[0.0002, 0.0018]$, and the sensitivity results on synthetic data $(\beta = 0.25)$ of $\sqrt{AMSE}$ are reported as follows.
>
> | Parameter $\gamma$ | $\sqrt{AMSE}$ |
> |--------------------|---------------|
> | 0.0002             | 0.1123        |
> | 0.0004             | 0.1111        |
> | 0.0006             | 0.1095        |
> | 0.0008             | 0.1089        |
> | 0.001               | 0.1074        |
> | 0.0012             | 0.1096        |
> | 0.0014             | 0.1108        |
> | 0.0016             | 0.1145        |
> | 0.0018             | 0.1156        |
>
> **Q4**: The efficiency of the method.
>
> **A4**: Running time results are given as follows.
> Although our method takes more time to solve optimal transport, our method achieves better performance compared with others.
>
> 1) Continuous treatment setting on synthetic data $(\beta = 0.25)$ in one realization
>
> | Methods  | Times |
> |----------|-------|
> | ORIC     | 135s  |
> | VCNet+TR | 23s   |
> | VCNet    | 17s   |
> | ADMIT    | 47s   |
> | ACFR     | 24s   |
> | DRNet    | 26s   |
> | GPS+MLP  | 25s   |
> | MLP      | 18s   |
> | GPS      | 9s    |
> | BART     | 7s    |
> | KNN      | 8s    |
>
> 2) Binary treatment setting on the IHDP-1000 data in one realization
>
> | Methods   | Times |
> |-----------|-------|
> | ORIC      | 76s   |
> | CFRNet    | 47s   |
> | DragonNet | 41s   |
> | DKLITE    | 4s    |
> | ESCFR     | 165s  |
> | CausalOT  | 4s    |
> | GANITE    | 4s    |
> | BART      | 0.2s  |
> | OLS       | 0.2s  |
> | KNN       | 0.3s  |

---

> > ### Author Response · Authors · 2024-11-25
> >
> > **Q5**: Limitations and future work.
> >
> > **A5**: Thank you for the valuable suggestions. The major limitations are discussed as follows.
> >
> > 1) Our approach of confounding bias reduction relies on the assumption of ignorability, which means that all the confounders are observed. In the future, we will further investigate the situation with unobserved confounders.
> >
> > 2) Our method involves multiple optimal transport problems, which is of high computational complexity. In the future, we will consider more efficient algorithms to solve the optimal transport problem.
> >
> > **Q6**: Regarding the potential outcome and the treatment effect.
> >
> > **A6**:
> >
> > 1) Thank you for the valuable suggestion. We will use $\mu$ to denote the potential outcome in the revision.
> >
> > 2) Since we aim to propose a unified framework for both binary and continuous treatments, we consider the potential outcome and AMSE, which are widely used for the continuous setting and can also be used for the binary setting.
> >
> > 3) Our theoretical results can be applied to PEHE. More details here.
> >
> > We use binary treatment as an example to illustrate that our theoretical results can be applied to PEHE.
> > Based on the assumptions in Theorem 2,
> > we first decompose PEHE for the true causal effect $\tau(x)=\mu_1(x)-\mu_0(x)$ as follows:
> >
> > \begin{align}
> > \varepsilon_{PEHE}
> > =E_{x\sim q(x)}[\ell(h_1(x) - h_0(x), \mu_1(x) - \mu_0(x))]
> > \leq E_{x\sim q(x)}[\ell(h_1(x), \mu_1(x))]
> > +E_{x\sim q(x)}[\ell(h_0(x), \mu_0(x))]
> > =\varepsilon_{q} (h_1) + \varepsilon_{q} (h_0)
> > \end{align}
> >
> > where $\ell$ is the $L_p$-norm based loss function and has the triangle inequality property.
> >
> > And We define the estimation error of the potential outcome function $\mu_1(x)$ and $\mu_0(x)$ in treatment and control groups, respectively:
> > \begin{align}
> > \varepsilon_{q_{1}} (h_1)=E_{x\sim q_{1}(x)}\ell (h_{1}(x) ,\mu_{1}(x))
> > \end{align}
> > \begin{align}
> > \varepsilon_{q_{0}} (h_0)=E_{x\sim q_{0}(x)}\ell ( h_{0}(x) ,\mu_{0}(x))
> > \end{align}
> >
> > According to Eq. (12), we have
> > \begin{align}
> > \varepsilon_{PEHE} \leq \varepsilon_{q} (h_1) + \varepsilon_{q} (h_0)
> > \leq \varepsilon_{q_1} (h_1) + \varepsilon_{q_0} (h_0) + \mathcal{W}(c, q_1, q) + \mathcal{W}(c, q_0, q)
> > \end{align}
> >
> > 4) For the experiments of binary treatments, we also report the results of PEHE, which demonstrate the effectiveness of our method.
> >
> >
> > **Q7**: Regarding the metric MISE.
> >
> > **A7**: As shown in [3], when considering continuous dosage without different kinds of treatments, the MISE criterion corresponds to the AMSE criterion mentioned. Therefore, we adopt AMSE in our submission.
> >
> > [1] Wang H, Fan J, Chen Z, et al. Optimal transport for treatment effect estimation[J]. Advances in Neural Information Processing Systems, 2024, 36.
> >
> > [2] Li Q, Wang Z, Liu S, et al. Causal optimal transport for treatment effect estimation[J]. IEEE transactions on neural networks and learning systems, 2021, 34(8): 4083-4095.
> >
> > [3] Kazemi A, Ester M. Adversarially balanced representation for continuous treatment effect estimation Proceedings of the AAAI Conference on Artificial Intelligence. 2024, 38(12): 13085-13093.

---

> > > ### Comment · Reviewer_E4XK · 2024-11-27
> > >
> > > Thank you for your detailed response and helpful changes! I have updated my score.

---

> > > > ### Author Response · Authors · 2024-11-28
> > > >
> > > > Thank you for the positive comments！

---

### Official Review · Reviewer_aEJu · 2024-10-31

**Soundness:** 3
**Presentation:** 2
**Contribution:** 2
**Rating:** 5
**Confidence:** 3

**Summary:**

This paper presents a novel approach for measuring the causal effect given a treatment. The model, called ORIC, is designed to directly predict outcomes rather than causal effects and is based on the formulation of optimal transport to reduce confounding bias in observational data. Unlike typical causal inference models that often split the data according to treatments and train separate models (e.g., T-learner), ORIC is designed to be effectively trained without data splitting, which is the most significant contribution of this paper.

**Strengths:**

While there have been a few instances in the literature over the past few years where the formulation of optimal transport has been borrowed to solve the problem of measuring causal effects, this paper presents yet another new perspective, which I find intriguing. The formulation that naturally combines representation learning through the $\phi$ function with it could be widely utilized in other related research in the future.

**Weaknesses:**

The paper lacks a convincing explanation or example of how avoiding data splitting is practically beneficial in certain cases. In the case of binary treatment, splitting into just two sub-populations means the sample size is halved. Still, it is hard to see how this significantly worsens the estimation of the conditional covariate distribution. It is unclear whether the assumption is that one population is extremely smaller than the other or if the issue is that both populations are too small, making splitting problematic. This paper aims to apply to observational data rather than RCTs, but in real-world data, while there may be confounding bias, the sample size is not too small and often large. Therefore, it needs to be more clearly explained in which scenarios this methodology is more beneficial.

The approach in this paper that avoids data splitting starts from equations (3) and (4), where AMSE is defined. It is necessary to explain the validity of why minimizing AMSE should be the goal and to provide more details on the mathematical properties of the results produced by minimizing AMSE. AMSE can be also expressed as follows: $AMSE=\int_T[\int_Xl(h_t,\tau_t)p(x|t)dx]p(t)^2 dt$. This includes the term $p(t)$ twice. Since $\epsilon_{q_t}(h_t) \equiv \int_X l(h_t,\tau_t)p(x|t)dx$ is biased, the inclusion of the $p(t)$ term is necessary. However, including it twice is not immediately convincing. For example, when defining PEHE, the $p(t)$ term is included only once: $PEHE=\int_T [\int_X ((h_1-\tau_1) – (h_0-\tau_0))^2  p(x|t)dx] p(t) dt$. Because AMSE includes $p(t)$ twice, through Theorem 3, the integral in equation (16) is calculated over the joint distribution $p(x,t)$ rather than $p(x|t)$, which enables to avoid data splitting. If the integral in equation (16) were calculated over $p(x|t)$, data splitting would still be required. This paper intentionally designed AMSE to avoid data splitting and aimed to reduce it through its upper bound. Therefore, it is necessary to explain why minimizing AMSE guarantees unbiased and accurate causal inference. As it stands, minimizing AMSE might lead to excessive bias depending on the data. For instance, does it sufficiently minimize $\epsilon_q(h_0)$ when $p(t=1) >> p(t=0)$? Additional proofs or demonstrations seem necessary.

**Questions:**

[1] Please provide a rebuttal or explanation for the points raised in the weaknesses section.

[2] Since the computational complexity of the Sinkhorn algorithm used to calculate optimal transport increases quadratically with the number of data points, the algorithm in this paper would have a significant computational load for large datasets. How does it compare to other algorithms?

[3] In Figure 1, the $\theta$ neural network is not depicted. How is the $\theta$ neural network implemented? The explanation of the $\theta$ model is insufficient, making the understanding of lines 360 to 364 somewhat ambiguous. How does the $\theta$ model depend on $t$?

[4] Can this paper be understood in the context of S-learner vs. T-learner? In other words, is this paper proposing an S-learner framework?

---

> ### Author Response · Authors · 2024-11-25
>
> Thank you for the valuable and insightful comments.
> We will revise the submission according to the
> comments. Our responses are as follows.
>
> **Q1**: Explanation regarding data splitting.
>
> **A1**:
>
> 1) Theoretically, more samples are helpful for better distribution estimation and alignment, which is supported by Eq. (15) in Theorem 3, in which the term $O(1/\sqrt{\delta n})$ indicates that more samples can induce a tighter upper bound.
>
> 2) In practice, less samples bring higher variances. This issue becomes even worse when multiple or continuous treatments are considered. More possible treatment values will induce fewer samples in each subset, resulting in higher variances and inaccurate distribution estimation and alignment. Therefore, it is important to improve the data efficiency to leverage more samples [1].
>
>
> **Q2**: Regarding AMSE, Eq. (16), and data splitting.
>
> **A2**:
>
> 1) AMSE is unbiased since the prediction loss is measured on unbiased distribution $q(x)$.
> AMSE is usually adopted as the target to minimize in causal inference for continuous treatments [2].
> Since all the combinations of $x$ and $t$ are considered and the integral is conducted on $q(x)p(t) dx dt$,
> $p(t=1) \gg p(t=0)$ does not affect AMSE as long as $p(t=0) > 0$, which is satisfied because of the positive assumption.
>
> 2) AMSE involves counterfactual outcomes and is intractable.
> Different from AMSE, the factual loss $\mathcal{L}$ in Eq. (16) is tractable but is conducted on the conditional distribution $q\_t(x) = p(x|t)$ rather than the marginal distribution $q(x)$. We propose to balance the distribution of $q\_t(x)$ and $q(x)$ by minimizing the Wasserstein distance between them, so that the model trained on balanced $q\_t(x)$ can be well generalized to $q(x)$. As shown in Theorem 3, minimizing the Wasserstein distances and $\mathcal{L}$ can reduce the upper bound of AMSE.
>
> 3) For the Wasserstein distance $\mathcal{W} (c, \hat{q}\_t, \hat{q})$ involving $\hat{q}\_t(x) = \hat{p}(x|t)$, we do not conduct data splitting. Instead of considering samples receiving $t$ only, we model $\hat{q}\_t = \sum_{i=1}^{n} \hat{q}\_t(x_i) \delta\_{x\_i}$ in Eq. (14), where $\hat{q}\_t(x\_i)$ is estiamted based on the generalized propensity scores. As a result, all the training samples are involved in the empirical distributions, which avoids the issue of
> data splitting and enhances the performance of distribution estimation.
>
>
> **Q3**: Computational complexity.
>
> **A3**: Running time results are given as follows.
> Although our method takes more time to solve optimal transport, our method achieves better performance compared with others.
>
> 1) Continuous treatment setting on synthetic data $(\beta = 0.25)$ in one realization
>
> | Methods  | Times |
> |----------|-------|
> | ORIC     | 135s  |
> | VCNet+TR | 23s   |
> | VCNet    | 17s   |
> | ADMIT    | 47s   |
> | ACFR     | 24s   |
> | DRNet    | 26s   |
> | GPS+MLP  | 25s   |
> | MLP      | 18s   |
> | GPS      | 9s    |
> | BART     | 7s    |
> | KNN      | 8s    |
>
> 2) Binary treatment setting on the IHDP-1000 data in one realization
>
> | Methods   | Times |
> |-----------|-------|
> | ORIC      | 76s   |
> | CFRNet    | 47s   |
> | DragonNet | 41s   |
> | DKLITE    | 4s    |
> | ESCFR     | 165s  |
> | CausalOT  | 4s    |
> | GANITE    | 4s    |
> | BART      | 0.2s  |
> | OLS       | 0.2s  |
> | KNN       | 0.3s  |
>
> **Q4**: Implementation of $\theta$
>
> **A4**: We are sorry for the confusion. The implementation of $\theta$ is based on [3] and described as follows. Assuming that the conditional distribution of treatment given covariates is Gaussian, i.e., $P(t \mid x_i) \sim \mathcal{N}(\theta(\phi(x_i)), \sigma^2)$. We can estimate the parameters by maximizing the likelihood:
>
> $\max_{\theta,\sigma} L(\hat{\theta},\hat{\sigma};t,x):=\prod_{i=1}^{n}  \frac{1}{\sqrt{2\pi \sigma^2}} \exp\left(-\frac{1}{2\sigma^2}(t_i - \theta(\phi(x_i)))^2\right)$.
>
> After that, the estimated generalized propensity score is given by:
>
> $\hat{p}(t \mid x_i) = \frac{1}{\sqrt{2\pi \hat{\sigma}^2}} \exp\left(-\frac{1}{2\hat{\sigma}^2}(t - \hat{\theta}(\phi(x_i)))^2\right)$.
>
> We will revise the submission to make it clear.

---

> > ### Author Response · Authors · 2024-11-25
> >
> > **Q5**: Regarding S-learner and T-learner.
> >
> > **A5**:
> >
> > 1) S-learner learns one model to predict potential outcomes, while T-learner learns multiple models to predict potential outcomes of different treatments separately. In this sense, our method can be regarded as a S-learner.
> >
> > 2) The focus of our submission is to avoid data splitting to improve data efficiency rather than the comparison between S-learner and T-learner. Even for S-learner, when considering distribution alignment between different groups receiving different treatments, reducing distribution shift after data splitting is still a common choice. Different from them, we propose to avoid data splitting and leverage more samples for learning.
> >
> >
> > [1] Kaddour J, Zhu Y, Liu Q, et al. Causal effect inference for structured treatments[J]. Advances in Neural Information Processing Systems, 2021, 34: 24841-24854.
> >
> > [2] Nie L, Ye M, Nicolae D. VCNet and Functional Targeted Regularization For Learning Causal Effects of Continuous Treatments. International Conference on Learning Representations.
> >
> > [3] Hirano K and Imbens GW. The propensity score with continuous treatments. In: Gelman A and Meng XL (eds) Applied bayesian modeling and causal inference from incomplete-data perspectives. Oxford, UK: Wiley, 2004, pp.73–84.

---

> > > ### Comment · Reviewer_aEJu · 2024-11-26
> > >
> > > Questions 3, 4, and 5 have been resolved with your answers. Thank you. However, I still have questions regarding questions 1 and 2.
> > >
> > > **Q 1:**
> > > Of course, the number of samples is important, and if it decreases, the uncertainty of the model increases. The fact that the term "data splitting" is included in the title indicates that it is the most important contribution of this paper. I want to gauge the practical significance of this contribution. For example, in a situation where only binary treatment (0 and 1) is considered, it is understood that the sample size is halved with existing methods and models, whereas this model can prevent the sample size from being halved. However, halving the sample size does not seem to be a major constraint. If there is a model that works properly with 1,000 samples, and we already have 2,000 samples, splitting them in half still gives us 1,000 samples. My question is whether the concept of data splitting is based on the number of treatments as I understand it, and if so, whether the effect of this paper is not significant for binary treatment. If so, what scenario would this methodology be most effective in?
> > >
> > > **Q 2:**
> > > I have reviewed the referenced paper [2]. I am a bit confused about whether the $\psi(t)$ used there is equivalent to equation (3) in the author's paper. Is there another paper that uses AMSE? If so, could you share it? What do you think about my comparison with PEHE above? PEHE and AMSE seem to have different properties. Even if $p(t=0)$ is positive, if it is very close to 0, AMSE only considers one side when $t=1$. In fact, PEHE can have $p(t)$ attached twice. Since the integrand is symmetric for binary treatment, having $p(t)$ attached twice still results in PEHE. On the other hand, if an integrand has $t$-dependence like the integrand of AMSE, attaching $p(t)$ once more may cause it to skew towards the side where $p(t)$ is larger. I am not 100% sure about this and believe you, as someone who has conducted the research and experiments, would know more. The process of calculating the upper bound of AMSE using rigorous and sophisticated mathematical methodology is beautiful. My concern is whether optimizing based on that AMSE ensures unbiased inference, and I want to have a solid understanding of this.

---

> > > > ### Author Response · Authors · 2024-11-28
> > > >
> > > > Thank you for the reply and valuable comments.
> > > > Our responses are as follows.
> > > >
> > > > **Q1**: Regarding data splitting.
> > > >
> > > > **A1**: Yes, data splitting is related to the number of treatment values. As the number of treatment values increases, the number of samples in each group decreases, and the issue of data splitting becomes more severe. For the binary treatment setting, the issue of data splitting is milder compared with the setting of more treatment values. While for the setting of continuous treatments, the issue becomes more severe since multiple discrete values of the treatment are considered, and each group only includes a small part of the samples, highly reducing the data efficiency and the precision of distribution estimation.
> > > >
> > > > **Q2**: Regarding $AMSE$ and $PEHE$.
> > > >
> > > > **A2**:
> > > >
> > > > 1) $\psi(t)$ in [2] is equivalent to $\int_{\mathcal{X}} \mu(x,t) p(x) dx$ of our submission, in which the potential outcome is revised as $\mu$ according to the comments of Reviewer E4XK.
> > > >
> > > > 2) $AMSE$ is also used in [a][b], in which $AMSE$ is called EMSE and MISE, respectively.
> > > >
> > > > 3) $AMSE$ is the expected loss measured on the marginal distribution $p(x)$. Following [a][b], $AMSE$ is defined as follows
> > > > \begin{align}
> > > > AMSE = \int_{\mathcal{T}} \int_{\mathcal{X}} \ell(h_t(x), \mu_t(x)) p(x) p(t) dx dt.
> > > > \end{align}
> > > > For $PEHE$, the generalized $PEHE$ compares the outcome of each treatment $t \neq 0$ with the outcome with $t=0$ and is given as follows [c]
> > > > \begin{align}
> > > > PEHE_g = \int_{\mathcal{T} \setminus \{0\}} \int_{\mathcal{X}}
> > > > \ell \big( h_t(x) - h_{0}(x), \mu_t(x) - \mu_{0}(x) \big)
> > > > p(x) p(t| t \neq 0) dx dt,
> > > > \end{align}
> > > > and the pairwise $PEHE$ compares the outcome of each treatment pair $(t, t')$ and is given as follows [d][e]
> > > > \begin{align}
> > > > PEHE_p = \int_{\mathcal{T} \times \mathcal{T}} \int_{\mathcal{X}}
> > > > \ell \big( h_t(x) - h_{t'}(x), \mu_t(x) - \mu_{t'}(x) \big)
> > > > p(x) p(t) p(t') dx dt dt'.
> > > > \end{align}
> > > >
> > > > From the above definitions, we observe that for the multiple-treatment setting, if some treatment value has a very small probability,
> > > > i.e. $p(t) \rightarrow 0$,
> > > > both the outcome in $AMSE$ and the causal effect in $PEHE$ related to $t$ are tended to be ignored.
> > > > In this sense, $AMSE$ and $PEHE$ have similar behaviors in the multiple-treatment setting.
> > > >
> > > > While for $PEHE$ in the binary setting, $\mathcal{T}$ includes only two values,
> > > > i.e. $0$ and $1$,
> > > > both $PEHE_g$ and $PEHE_p$ degenerate to the following form
> > > > \begin{align}
> > > > PEHE_b = \int_{\mathcal{X}}
> > > > \ell \big( h_1(x) - h_{0}(x), \mu_1(x) - \mu_{0}(x) \big)
> > > > p(x) dx.
> > > > \end{align}
> > > > As there is only one combination of different treatment values,
> > > > the issue caused by a small $p(t)$ does not exist.
> > > > In this situation, $AMSE$ and $PEHE$ indeed have different properties.
> > > >
> > > > Since we consider a unified framework for both binary and continuous settings, we mainly focus on $AMSE$ in the submission.
> > > > According to A6 to Reviewer E4XK,
> > > > for the binary setting, $AMSE$ is an upper bound of $PEHE$, and our theoretical analysis based on $AMSE$ can be easily extended to $PEHE$.
> > > >
> > > > This is indeed an interesting question, which prompts deep thinking and enhances the understanding of $AMSE$ and $PEHE$.
> > > >
> > > > 4) In causal inference, confounding bias brings estimation bias of potential outcomes,
> > > > which means that the loss on the observed distributions $p(x|t)$ is a biased estimate of the desired population loss on the marginal distribution $p(x)$,
> > > > and the model minimizing the loss on subsets cannot be well generalized to the whole population [f].
> > > > By minimizing $AMSE$, the expected loss between $h_t(x)$ and $\mu_t(x)$ is minimized on the marginal distribution $p(x)$, which is independent with $t$ without confounding bias.
> > > >
> > > > [a] Wang X, Lyu S, Wu X, et al. Generalization bounds for estimating causal effects of continuous treatments. Advances in Neural Information Processing Systems, 2022, 35: 8605-8617.
> > > >
> > > > [b] Kazemi A, Ester M. Adversarially balanced representation for continuous treatment effect estimation. Proceedings of the AAAI Conference on Artificial Intelligence. 2024, 38(12): 13085-13093.
> > > >
> > > > [c] Guo X, Zhang Y, Wang J, Long M. Estimating heterogeneous treatment effects: Mutual information bounds and learning algorithms. In International Conference on Machine Learning 2023.
> > > >
> > > > [d] Schwab, P., Linhardt, L., and Karlen, W. Perfect Match:
> > > > A Simple Method for Learning Representations For
> > > > Counterfactual Inference With Neural Networks. arXiv
> > > > preprint arXiv:1810.00656, 2019.
> > > >
> > > > [e] Kaddour, J., Zhu, Y., Liu, Q., Kusner, M. J., and Silva, R.
> > > > Causal effect inference for structured treatments. Advances
> > > > in Neural Information Processing Systems, 34:
> > > > 24841–24854, 2021.
> > > >
> > > > [f] Johansson FD, Shalit U, Kallus N, Sontag D. Generalization bounds and representation learning for estimation of potential outcomes and causal effects. Journal of Machine Learning Research. 2022;23(166):1-50.

---

> > > > > ### Comment · Reviewer_aEJu · 2024-12-02
> > > > >
> > > > > You provided good answers to my questions. However, my remaining concern is that while PEHE is symmetric with respect to $p(t)$, AMSE is not. In other words, if $p(t)$ is heavily skewed, it is still unclear what results AMSE will produce. Therefore, I will raise my score to 5 instead of 6 or 8. I fundamentally think this is a good paper. If it gets accepted, I suggest including an example in the appendix showing that AMSE provides good results without bias even when $p(t)$ is heavily skewed. If not, this paper might be understood as having a flaw in general application.

---

> > > > > > ### Author Response · Authors · 2024-12-04
> > > > > >
> > > > > > Thank you for the reply and valuable comments. Our responses are as follows.
> > > > > >
> > > > > > It is difficult to consider a situation where $p(t)$ is heavily skewed.
> > > > > > Instead, we approximate skewed $p(t)$ by setting the $n_1 \gg n_0$ or $n_1 \ll n_0$,
> > > > > > where $n_1$ and $n_0$ are the numbers of samples in the treated and control groups, respectively.
> > > > > > We remove some samples to obtain skewed $n_1$ and $n_0$.
> > > > > > The results on the News data are reported as follows,
> > > > > > where $t:c$ means $n_1 : n_0$.
> > > > > > Our method consistently achieves promising performance.
> > > > > >
> > > > > >
> > > > > > | t:c | 1:10              | 1:10              | 1:10              | 1:5               | 1:5               | 1:5               | 10:1                | 10:1               | 10:1                | 5:1               | 5:1               | 5:1               |
> > > > > > |-------------------|-------------------|-------------------|-------------------|-------------------|-------------------|-------------------|---------------------|--------------------|---------------------|-------------------|-------------------|-------------------|
> > > > > > | metric   | $\sqrt{PEHE}$     | $MAE$             | $\sqrt{AMSE}$     | $\sqrt{PEHE}$     | $MAE$             | $\sqrt{AMSE}$     | $\sqrt{PEHE}$       | $MAE$              | $\sqrt{AMSE}$       | $\sqrt{PEHE}$     | $MAE$     |$\sqrt{AMSE}$           |
> > > > > > | CFRNet            | $6.154 \pm 5.621$ | $3.304 \pm 5.358$ | $6.705 \pm 4.470$ | $6.104 \pm 4.934$ | $3.155 \pm 3.365$ | $6.665 \pm 4.309$ | $11.839 \pm 15.701$ | $6.915 \pm 14.161$ | $12.844 \pm 15.884$ | $4.614 \pm 2.925$ | $1.878 \pm 1.581$ | $5.209 \pm 2.856$ |
> > > > > > | ORIC              | $2.094 \pm 0.555$ | $0.673 \pm 0.639$ | $2.357 \pm 0.564$ | $1.867 \pm 0.487$ | $0.320 \pm 0.286$ | $2.119 \pm 0.503$ | $2.292 \pm 0.678$   | $0.554 \pm 0.345$  | $2.609 \pm 0.691$   | $2.106 \pm 0.594$ | $0.184 \pm 0.157$ | $2.383 \pm 0.607$ |
> > > > > >
> > > > > > Besides, for the ihdp data, $n_1 : n_0 = 139:608$ is also skewed,
> > > > > > and our method also performs well.

---

### Official Review · Reviewer_3uWP · 2024-11-01

**Soundness:** 3
**Presentation:** 3
**Contribution:** 2
**Rating:** 5
**Confidence:** 4

**Summary:**

This work mainly studies the optimal transport method to reduce bias in causal inference. The authors propose a distribution alignment paradigm that leverages all training samples, avoiding the need for data splitting, which is a common limitation of existing methods. The theoretical contributions and empirical results demonstrate the effectiveness of the proposed method in both binary and continuous treatment settings.

**Strengths:**

1. The paper is well-structured and flows naturally.

2. This paper proposes a balanced  algorithm is designed that can effectively reduce confounding bias without data splitting.

3. The experimental studies are well done. A sufficient amount of empirical evidence for the proposed method is provided.

**Weaknesses:**

1. Theorem 2 holds under the assumption that $\mathcal{H}$ is an RKHS. Given this, it’s unclear why the author didn't opt to use kernel methods for learning representations, as they seem more aligned with the conditions of the theorem.

2. While the experimental results show notable improvements, it's not entirely clear why the method performs so well. The theoretical bounds appear loose, so it would be valuable to explore why the method is still effective in practice. Gaining a deeper understanding would make the approach more reliable.

3. It would be beneficial to conduct ablation studies on the loss function involving Wasserstein distances. This could help identify which components contribute most to the performance gains and provide further insight into the tightness of the theoretical bounds.

4. What are the key advantages and differences between the author's debiasing method and other re-weighting approaches? Clarifying this comparison would be helpful.

**Questions:**

1. The algorithm relies on the ignorability assumption for its validity. I'm curious about its robustness in the presence of unobserved confounders. Could it still perform well under such circumstances?

2. If $X$ is a high-dimensional variable, how to estimate the probability mass $q(x_i)$? I worry about the curse of dimensionality


I will raise the score if these items are addressed.

---

> ### Author Response · Authors · 2024-11-25
>
> Thank you for the valuable and insightful comments.
> We will revise the submission according to the
> comments. Our responses are as follows.
>
> **Q1**: Regarding the assumption of RKHS.
>
> **A1**:
>
> 1) The kernel method requires to calculate kernel functions between all the pairs of two samples, which has high computational complexity.
>
> 2) The kernel function between two samples $x$ and $x'$ is the inner product of their mapped representations, i.e., $k(x, x') = \langle \phi(x), \phi(x') \rangle$. In practice, we employ a neural network to approximate the mapping function $\phi$. A similar approach has also been used in [1].
>
> **Q2**: Explanation of the performance improvement.
>
> **A2**: The performance improvement mainly comes from two perspectives.
>
> 1) Existing distribution alignment methods usually split samples into multiple subsets, each of which only includes the samples receiving a specific treatment. Different from them, we do not split data into smaller subsets. This means that more samples are leveraged for learning. This advantage is theoretically supported by Eq. (15) in Theorem 3, in which the term $O(1/\sqrt{\delta n})$ indicates that more samples induce a tighter upper bound.
>
> 2) For balanced representation learning achieved by minimizing the Wasserstein distances, we further introduce the generalized propensity scores into the model. This approach can adaptively assign weights for samples, making samples with higher generalized propensity scores contribute more to the model.
>
> **Q3**: Ablation study on the loss function involving Wasserstein distances.
>
> **A3**: We have conducted the ablation study on the loss function involving Wasserstein distances in the revision. The results are reported as follows.
>
> | Methods | Synthetic$(\beta = 0.25)$ | Synthetic$(\beta = 0.5)$ | Synthetic$(\beta = 0.75)$ | Synthetic$(\beta = 1)$ | IHDP                | News                |
> |---------|---------------------------|--------------------------|---------------------------|------------------------|---------------------|---------------------|
> | ORIC    | 0.1098 $\pm$ 0.0273       | 0.1234 $\pm$ 0.0388      | 0.1313 $\pm$ 0.0464       | 0.1168 $\pm$ 0.0316    | 0.3595 $\pm$ 0.0304 | 0.1507 $\pm$ 0.0406 |
> | ORIC without wass | 0.2077 $\pm$ 0.0238       | 0.2028 $\pm$ 0.0203      | 0.2022 $\pm$ 0.0210       | 0.2161 $\pm$ 0.0157    | 0.6303 $\pm$ 0.0826 | 0.4255 $\pm$ 0.2115 |
> | ORIC without wass and GPS | 0.2083 $\pm$ 0.0275       | 0.2042 $\pm$ 0.0311      | 0.2044 $\pm$ 0.0252       | 0.2044 $\pm$ 0.0252    | 0.6566 $\pm$ 0.0710 | 0.4355 $\pm$ 0.2098 |
>
> **Q4**: Comparison with other re-weighting approaches.
>
> **A4**:
>
> 1) Propensity score-based methods employ propensity scores to re-weight samples. Different from them, we leverage propensity scores to model the conditional distributions, and incorporate propensity scores into the optimal transport model for balanced representation learning.
>
> 2) Some re-weighting methods learn weights for samples to balance the distributions of different groups, where the distribution shift is measured by a predefined metric. However, for distributions with a very large shift, re-weighting cannot well reduce the distribution shift since the support sets are fixed. Different from them, we characterize the confounding bias by considering the balancing error between the conditional distribution and the marginal distribution as shown in Eqs. (6) and (7),
> and connect the confounding bias with the optimal transport, which motivates us to learn balanced representations to reduce the confounding bias.
>
> **Q5**: Regarding unobserved confounders.
>
> **A5**: Since we characterize the confounding bias by measuring the discrepancy between $q_t(x)$ and $q(x)$, the ignorability assumption is required. If unobserved confounders exist, the confounding bias can not be fully captured by considering $q_t(x)$ and $q(x)$ only. We will investigate the situation with unobserved confounders in the future.
>
> **Q6**: How to estimate the probability mass $q(x_i)$ for high-dimensional variables?
>
> **A6**: The probability mass $q_t(x)$ is estimated by generalized propensity scores, which can be estimated by a regression model. For high-dimensional variables, we can reduce the dimension by PCA or neural networks, or introduce the $\ell_1$ norm to select informative variables.
>
> [1] Shalit U, Johansson F D, Sontag D. Estimating individual treatment effect: generalization bounds and algorithms. International conference on machine learning. PMLR, 2017: 3076-3085.

---

### Official Review · Reviewer_pqUy · 2024-11-02

**Soundness:** 3
**Presentation:** 2
**Contribution:** 2
**Rating:** 6
**Confidence:** 3

**Summary:**

The paper suggests the use of a novel estimator of conditional average treatment effects and related causal objects when there is confounding explained by observable covariates. The method can be adjusted to allow for continuous treatments as well as discrete treatments. The proposed methods employ a particular adjustment for covariate imbalance that is based on an optimal transport. The authors also propose a related measure of covariate imbalance which they suggest may be used to assess the threat of confounding bias. The authors benchmark their methods against alternative approaches on established synthetic datasets.

**Strengths:**

The use of optimal transport methods to adjust for covariate imbalance seems to me a promising area of research. The performance of the proposed methods on synthetic data is very encouraging.

**Weaknesses:**

I found the motivation for the methods unclear. The authors point out that the difference between the feasible loss function $\int\varepsilon_{q_{t}}(h_{t})p(t)dt$ and the infeasible loss function $\int\varepsilon_{q}(h_{t})p(t)dt$ can be bounded by the Wasserstein distance between $q_{t}$ and $q$. It does not follow immediately that minimizing the sum of the feasible loss and this distance metric should lead to superior estimation performance. Indeed, the objective in equation (20) is minimized over three parameters $\phi$, $\psi$, and $\theta$, but only $\phi$ is shared between the feasible loss estimate $\hat{\mathcal{L}}$ and the Wasserstein metric. The estimate of $\tau_{t}(x)$ depends only on $\phi$ and $\psi$. Thus the benefits of including this Wasserstein distance in the optimization problem must result only from improved feature learning (i.e., an improved choice for $\phi$). In fact, I was rather perplexed that the loss $\hat{\mathcal{L}}$ did not incorporate the generalized propensity score $\theta$. It seemed to me from earlier parts of the paper that calculating the generalized propensity score by optimal transport, and using this to recover covariate balance, was likely to be the main point of the paper. But unless there is an error in the description of the methods, then $\theta$ is not directly used in learning $\tau_{t}(x)$, only impacting that problem through its impact on the choice of $\phi$. Incidentally, this in itself is rather complicated because the cost function $c_\phi(\cdot,\cdot)$ in the optimal transport is based on $\phi$, and setting $\phi$ to be identically zero would trivially make the Wassersteuin metric equal to zero.

Perhaps I am missing something here, particularly given that the very good performance on the benchmarking data, but I spent rather a long time trying to understand this and so, at the very least, I do not think it is well-explained.

I also found the description of the algorithm confusing. The algorithm contains a loop whose final step is to minimize the objective in (20) and the loop is iterated until convergence. But if we minimize this objective, then what changes in each iteration of the loop?

Finally, I find the description of the existing literature to be somewhat narrow. Causal inference methods have been developed over many decades and ML methods that incorporate data-splitting represent only a very recent and thin stratum of this much broader literature. In my opinion the authors ought to clarify when they are talking about causal inference methods as a whole, and when they are referring specifically to only recent causal inference methods from the ML literature.

**Questions:**

Why does the loss $\hat{\mathcal{L}}$ not incorporate the generalized propensity score? Give that it does not, what is the purpose of the Wasserstein part of the loss? Does it play a role other than improving the choice of $\phi$? If its only purpose is indeed to improve $\phi$ why should it achieve this? I am very curious to better understand the methods, particularly given the apparent superior performance in Section 5, but until I am able to grasp how/why the proposed methods  might work I do not feel I can recommend accepting the paper.

I wondered how GPS (without MLP) might perform on the data, and likewise for some other classical or non neural-network-based methods?

---

> ### Author Response · Authors · 2024-11-25
>
> Thank you for the valuable and insightful comments.
> We will revise the submission according to the
> comments. Our responses are as follows.
>
> **Q1**: Explanation of the method.
>
> **A1**:
>
> 1) Following [1], we aim to reduce AMSE that is the unbiased outcome estimation error for all the treatment values. Based on Lines 289-296 and Eq. (12), AMSE is upper-bounded by the Wasserstein distances between $q_t$ and $q$, as well as the feasible biased loss $\varepsilon_{q_t}(h_t)$. This analysis motivates us to minimize the upper bound of AMSE, i.e., the sum of the feasible loss and the Wasserstein distances, and design the algorithm in Section 4.3.
>
> 2) Generalized propensity scores obtained by $\theta$ are modeled as $\hat{q}_t$ and are used in the minimization of the Wasserstein distances, as shown in Eqs. (18) and (19), which tends to obtain balanced representations by learning $\phi$.
>
> 3) Without balanced representations, the outcome prediction model $\psi$ will be trained on biased data distributions, which means that the prediction model $\phi(\cdot,t)$ could fit the conditional distribution $q_t$ (i.e., samples receiving the treatment $t$) while not being well generalized to other subsets (i.e., samples receiving $t' \neq t$). By minimizing the Wasserstein distances based on $\phi$ and generalized propensity scores, we can obtain balanced representations, and $\phi$ can be well generalized to all the samples rather than a subset receiving specific treatments.
>
> 4) Setting $\phi$ to be identically zero will make the Wasserstein distance zero. On the other hand, setting $\phi$ to zero will induce a very large outcome prediction loss $\hat{\mathcal{L}}$ in Eq. (17). Therefore, minimizing $\hat{\mathcal{L}}$ can avoid the situation where $\phi$ being zero. This is discussed in Lines 290-296, which motivates us to propose Eq. (12).
>
> **Q2**: Algorithm description.
>
> **A2**: We are sorry for the confusion. Since Problem (20) involves multiple blocks of variables to optimize, we cannot minimize the objective in one iteration. Instead, in Step 9, we update the parameters of the model based on the gradient descent method. We will revise the submission to make it clear.
>
> **Q3**: Description of the existing literature.
>
> **A3**: Thank you for the valuable comments. In this paper, we mainly focus on the machine learning method for causal inference. We will revise the submission accordingly.
>
> **Q4**: The performance of classical or non neural-network methods.
>
> **A4**: We have conducted non neural-network methods in the revision. The results are reported as follows.
>
> | Methods | Synthetic$(\beta = 0.25)$  | Synthetic$(\beta = 0.5)$ | Synthetic$(\beta = 0.75)$ | Synthetic$(\beta = 1)$ | IHDP                | News                |
> |---------|----------------------------|--------------------------|---------------------------|------------------------|---------------------|---------------------|
> | ORIC    | 0.1098 $\pm$ 0.0273        | 0.1234 $\pm$ 0.0388      | 0.1313 $\pm$ 0.0464       | 0.1168 $\pm$ 0.0316    | 0.3595 $\pm$ 0.0304 | 0.1507 $\pm$ 0.0406 |
> | KNN     | 0.2339 $\pm$ 0.0294        | 0.2234 $\pm$ 0.0296      | 0.2211 $\pm$ 0.0235       | 0.2361 $\pm$ 0.0209    | 0.8364 $\pm$ 0.0917 | 0.6104 $\pm$ 0.4117 |
> | BART    | 0.2205 $\pm$ 0.0248        | 0.2108 $\pm$ 0.0312      | 0.2177 $\pm$ 0.0259       | 0.2238 $\pm$ 0.0212    | 0.6825 $\pm$ 0.0715 | 0.5639 $\pm$ 0.3125 |
> | GPS     | 0.2103 $\pm$ 0.0319        | 0.2056 $\pm$ 0.0345      | 0.2063 $\pm$ 0.0264       | 0.2219 $\pm$ 0.0238    | 0.7247 $\pm$ 0.0582 | 0.4422 $\pm$ 0.2033 |
>
> **Q5**: The reason why loss $\mathcal{\hat{L}}$ does not incorporate the generalized propensity score $\theta$.
>
> **A5**: Generalized propensity scores obtained by $\theta$ are modeled as $\hat{q}_t$ and are used in the minimization of the Wasserstein distances, as shown in Eq.s (18) and (19), which tends to obtain balanced representations by learning $\phi$.
> Based on balanced representations, the outcome prediction model $\psi$ can be trained on unbiased data distributions, making the prediction model $\phi(\cdot,t)$ generalized to all the samples rather than only the subset receiving the treatment $t$.
>
> [1] Nie L, Ye M, Nicolae D. VCNet and Functional Targeted Regularization For Learning Causal Effects of Continuous Treatments. International Conference on Learning Representations.

---

> > ### Comment · Reviewer_pqUy · 2024-11-28
> >
> > Thank you for your response. Regarding the motivation for the methods, could you perhaps elaborate on point 3: What does $\phi$ really represent here? What would be the oracle choice for $\phi$? Why does the Wasserstein part of the objective help you obtain a good choice for $\phi$?

---

> > > ### Author Response · Authors · 2024-12-01
> > >
> > > Thank you for the reply and valuable comments. Our responses are as follows.
> > >
> > > **Q1**:
> > > What does $\phi$ really represent here?
> > >
> > > **A1**:
> > > $\phi$ is a feature mapping function implemented by a neural network. $\phi(x)$ is the representation vector of the sample $x$ in a learned feature space.
> > >
> > > **Q2**:
> > > What would be the oracle choice for $\phi$?
> > >
> > > **A2**:
> > > The oracle choice for $\phi$ is a feature mapping function whose output representation vectors $\{ \phi(x) \}$ own the two properties:
> > >
> > > i) the outcome prediction loss $\hat{\mathcal{L}}$ in Eq. (17) is minimized, which means that the outcome information is well captured in $\phi(x)$.
> > >
> > > ii) the Wasserstein distances $\mathcal{W}(c_{\phi}, \hat{q}_t, \hat{q})$ is minimized, which means that the empirical distributions $\hat{q}_t$ an $\hat{q}$ defined in Eq. (14) are close to each other. As a result, the conditional distribution $\hat{q}_t$ and $\hat{q}$ are similar, and the outcome prediction model trained on the distribution $\hat{q}_t$ can be well generalized to the marginal distribution $\hat{q}$.
> > >
> > > **Q3**:
> > > Why does the Wasserstein part of the objective help you obtain a good choice for $\phi$?
> > >
> > > **A3**:
> > >
> > > 1) By minimizing the Wasserstein distances, we can learn balanced representations to reduce the confounding bias.
> > > Here, the balanced representations are obtained by the learned mapping function $\phi$.
> > >
> > > 2) According to Line 344 and Eq. (18), the Wasserstein distance is based on the underlying cost function $c_{\phi}(x_i, x_j) = || \phi(x_i) - \phi(x_j) ||$.
> > > A good $\phi$ is the one to minimize the Wasserstein distances $\mathcal{W}(c_{\phi}, \hat{q}_t, \hat{q})$ in Eq. (18), in which the optimal transport plan $\tilde{\pi}^t$ is obtained by solving Problem (19).
> > > In practice, given a mapping function $\phi$ parameterized by a neural network, we solve Problem (19) to obtain the optimal transport plan $\tilde{\pi}^t$.
> > > After that, we calculate the objective function (i.e., the Wasserstein distance in Eq. (18)) based on $\tilde{\pi}^t$,
> > > and then obtain the gradient of the objective with respect to the parameters of $\phi$,
> > > so that a gradient descent algorithm can be applied to update the parameters of $\phi$.
> > > A similar approach to calculate the Wasserstein distance and learn the corresponding feature mapping function is also used in [a].
> > >
> > > [a] Shalit, Uri, Fredrik D. Johansson, and David Sontag. "Estimating individual treatment effect: generalization bounds and algorithms." International conference on machine learning. PMLR, 2017.

---

> > > > ### Comment · Reviewer_pqUy · 2024-12-01
> > > >
> > > > Perhaps I am still missing something, but I still do not see how the fact that the cost function is defined using $\phi$ should help adjust for confounding. It seems that this feature of the estimator should mean that after applying $\phi$ to $x_i$, its conditional distribution given $t_i$ should not depend too much on $t_i$ (in the sense that the distribution of $\phi(x_i)$ conditional on $t_i=0$ is close in the Wasserstein metric to its distribution conditional on $t_i=1$), is that right? But why does that help with causal inference? What are the advantages of controlling for features $\phi(x_i)$ that are not too strongly dependent on $t_i$? I apologize for belaboring this point, but I really want to understand what drives the performance of the estimator.

---

> > > > > ### Author Response · Authors · 2024-12-02
> > > > >
> > > > > Thank you for the reply and valuable comments. Our responses are as follows.
> > > > >
> > > > > For simplicity, we answer the question by considering the setting of the binary treatments in the following.
> > > > > Our conclusion can be easily extended into the setting of continuous treatments.
> > > > >
> > > > > 1) Yes, the distribution of $\phi(x_i)$ conditional on $t_i = 0$ is close to the distribution conditional on $t_i = 1$ in the Wasserstein metric, which means that $\phi(x_i)$ is independent on $t_i$.
> > > > >
> > > > > 2) According to Theorem 1, the confounding bias characterized by the balancing error can be upper bounded by the Wasserstein distances.
> > > > > Motivated by this, we parameterize the Wasserstein distances by the feature mapping function $\phi$ and minimize the distances to reduce the confounding bias.
> > > > >
> > > > > 3) According to Eq. (12) and the discussion in Lines 288-295, we reduce AMSE by minimizing its upper bound, i.e., the factual loss and the Wasserstein distances.
> > > > > In other words, by minimizing the Wasserstein distances based on the feature mapping function $\phi$, the confounding bias is reduced, and the factual loss $\int_{\mathcal{T}} \varepsilon_{q_t} (h_t) p(t) dt$ is close to our target $AMSE$.
> > > > >
> > > > > 4) Intuitively, the confounding bias makes $q_1(x) \neq q_0(x)$, where the treatment assignment is affected by the covariates $x$.
> > > > > As a result,
> > > > > $\phi(x_i)$ is dependent on $t_i$,
> > > > > and the outcome prediction model $\psi_1$ trained on $q_1(x)$ cannot achieve good performance on $q_0(x)$.
> > > > > For example, if serious patients tend to receive better treatment ($q_1(x)$),
> > > > > while mild patients tend to receive a modest treatment ($q_0(x)$),
> > > > > then $\psi_1$ trained on serious patients cannot be generalized to mild patients to predict their outcomes receiving $t=1$,
> > > > > since the two groups follow significantly different distributions.
> > > > > By learning a $\phi$ to make $\phi(x_i)$ not dependent on $t_i$,
> > > > > we enforce the two groups follow a similar distribution,
> > > > > so that $\psi_1$ trained on $\phi(x_i)$ of serious patients can be well generalized to $\phi(x_i)$ of the wild patients.
> > > > > At the same time,
> > > > > by minimizing the factual loss in Eq. (17),
> > > > > we make sure that the potential outcome information is well preserved in $\phi$.

---

> > > > > > ### Comment · Reviewer_pqUy · 2024-12-02
> > > > > >
> > > > > > Thank you for your response. I feel I better understand your motivation, but unfortunately, I am not convinced that your reasoning works. It is indeed true that the confounding bias can be upper bounded by the difference in the conditional distribution of $x_i$ under $t_i=0$ and $t_i=1$. But this relies crucially on the fact that conditional on $x_i$, treatment is independent of potential outcomes. $\phi(x_i)$ is typically of lower dimension to $x_i$ and so there is no guarantee that conditional on $\phi(x_i)$, treatment will still be independent of potential outcomes. In fact, the very fact that $\phi(x_i)$ is chosen so that its conditional distribution does not depend strongly on treatment, means that using $\phi(x_i)$ is likely to be problematic. When we adjust for confounding, we aim to control for observables that can explain the association between $t_i$ and potential outcomes. Such variables must be correlated with $t_i$. By choosing $\phi$ so that $\phi(x_i)$ is not strongly related to $t_i$, this would seem to select $\phi(x_i)$ that are not helpful for adjusting for confounding.

---

> > > > > > > ### Author Response · Authors · 2024-12-04
> > > > > > >
> > > > > > > Thank you for the reply and valuable comments. Our responses are as follows.
> > > > > > >
> > > > > > > According to the ignorability Assumption,
> > > > > > > $ X $ is the confounders that simultaneously affect both $ T $ and $ Y $,
> > > > > > > which need to be adjusted to avoid confounding bias through the backdoor path $ T \leftarrow X \rightarrow Y $.
> > > > > > > Learning balanced representation $ \phi(X) $ is one way to adjust for the confounders. Specifically, by implementing the factual loss and Wasserstein distance together,
> > > > > > > we ensure that $ \phi(X) $ is related to $ Y $ while remaining independent of $ T $.
> > > > > > > From the perspective of causal graphs,
> > > > > > > this means that $\phi(X)$ is the mediator on the edge $X \rightarrow Y$, i.e., $X \rightarrow  \phi(X) \rightarrow Y$,
> > > > > > > and $\phi(X)$ blocks the backdoor path from $T$ to $Y$.
> > > > > > > Therefore,
> > > > > > > when we use balanced $\phi(X)$ and $T$ to train the model,
> > > > > > > we can approximately think of it as training on RCT data, avoiding confounding bias.
> > > > > > > Actually,
> > > > > > > the balanced representation-based method is already a classic paradigm for causal effect estimation, which is widely used in the machine learning community, e.g., [1-4]
> > > > > > >
> > > > > > > From a broader perspective,
> > > > > > > propensity score and prognostic score are two forms of dimensionality reduction for confounders.
> > > > > > > The propensity score is a mediator on the path $ X \rightarrow T $,
> > > > > > > while the prognostic score is a mediator on the path $ X \rightarrow Y $.
> > > > > > > According to [5] and [6],
> > > > > > > controlling any one of them is sufficient for causal inference.
> > > > > > > And as stated in the related work in [2], our balanced representation $ \phi(X) $ can be viewed as a special case of a prognostic score.
> > > > > > >
> > > > > > > [1] Shalit U, Johansson F D, Sontag D. Estimating individual treatment effect: generalization bounds and algorithms[C]//International conference on machine learning. PMLR, 2017: 3076-3085.
> > > > > > >
> > > > > > > [2] Johansson F D, Shalit U, Kallus N, et al. Generalization bounds and representation learning for estimation of potential outcomes and causal effects[J]. Journal of Machine Learning Research, 2022, 23(166): 1-50.
> > > > > > >
> > > > > > > [3] Kazemi A, Ester M. Adversarially balanced representation for continuous treatment effect estimation[C]//Proceedings of the AAAI Conference on Artificial Intelligence. 2024, 38(12): 13085-13093.
> > > > > > >
> > > > > > > [4] Guo X, Zhang Y, Wang J, et al. Estimating heterogeneous treatment effects: Mutual information bounds and learning algorithms[C]//International Conference on Machine Learning. PMLR, 2023: 12108-12121.
> > > > > > >
> > > > > > > [5] Rosenbaum P R, Rubin D B. The central role of the propensity score in observational studies for causal effects[J]. Biometrika, 1983, 70(1): 41-55.
> > > > > > >
> > > > > > > [6] Hansen B B. The prognostic analogue of the propensity score[J]. Biometrika, 2008, 95(2): 481-488.

---

### Meta-Review · Area_Chair_nrvn · 2024-12-17

**Metareview:**

Although this paper contains some potentially interesting ideas about correcting for the effect of confounding variables, in its current form, this paper has just too many weaknesses.  To me, the most severe point of criticism concerns the concept of correcting for confounders using variables that are (almost) uncorrelated with the treatment, which seems to be a serious conceptual problem, because the very nature of a confounder is that it jointly influences both treatment and outcome. Unfortunately, this conceptual problem could not be addressed in a clear way during the rebuttal and discussion phase.

**Additional Comments On Reviewer Discussion:**

The main this conceptual problem -- namely the unclear role of variables that are uncorrelated with treatment for correcting for confounding -- could not be addressed in a convincing way during the rebuttal and discussion phase.

---

### Decision · Program_Chairs · 2025-01-22

Reject